# Review on Stochastic Approach to Inflation

Diego Cruces [1,2] 

[1] Institut de Ciencies del Cosmos (ICCUB), Universitat de Barcelona, Martí i Franquès 1, 08028 Barcelona, Spain; dcruces@ub.edu; Tel.: +34-93-402-0817

[2] Departament de Física Quàntica i Astrofísica, Faculty of Physics, Universitat de Barcelona, Martí i Franquès 1, 08028 Barcelona, Spain

**Abstract:** We present a review on the state-of-the-art of the mathematical framework known as stochastic inflation, paying special attention to its derivation, and giving references for the readers interested in results coming from the application of the stochastic framework to different inflationary scenarios, especially to those of interest for primordial black hole formation. During the derivation of the stochastic formalism, we will emphasise two aspects in particular: the difference between the separate universe approach and the true long wavelength limit of scalar inhomogeneities and the generically non-Markovian nature of the noises that appear in the stochastic equations.

**Keywords:** stochastic inflation; primordial black holes; early universe; inflation



## 1. Introduction

Although cosmological inflation was first introduced as a possible solution to the hot Big Bang model problems [1,2], the study of vacuum fluctuations during this accelerated expansion of the universe also gives an explanation to the anisotropies observed in the Cosmic Microwave Background (CMB). The idea is that the different scales of fluctuations leave the observable universe during inflation, long after it, they reenter the observable universe at different times, being the scales reentering the cosmological horizon (horizon from now on) during the time of recombination the responsible for the CMB anisotropies. In fact, the almost gaussianity and scale invariance of the vacuum quantum fluctuations predicted by Slow Roll (SR) inflation has been remarkably well confirmed by the recent Plank satellite mission [3,4].

It is however true that, although inflation provides a causal mechanism to generate CMB anisotropies, the CMB is only accessible to a restricted range of scales, constraining the inflationary phase only during a limited time interval. In order to be accessible to smaller scales of inflation, that reenter the horizon before the recombination epoch, we must then seek for any other hint that help us to extend the time interval of inflation that we can constraint. Primordial Black Holes (PBHs) is one of those hints [5]. In addition to the fact that PBHs can probe the missing scales of inflation, they represent natural candidates not only for dark matter (DM) [6], but also as the seed of supermassive BHs at the center of massive galaxies [7] and even as the progenitors of some events that radiate the gravitational waves detected by the LIGO/VIRGO collaboration [8].

PBHs are expected to form if the amplitude of density perturbations from inflation is big enough such that when they reenter the horizon they will collapse into a Black Hole (BH). This is not the case for CMB scales, for which the SR approximation holds and where the amplitude of inhomogeneities is too small to form PBHs; however, for smaller scales, this amplitude is less constrained and it could grow until PBHs are possible to form, in this case, SR is not a good approximation and other inflationary regimes arise such as Ultra Slow Roll (USR) and Constant Roll (CR).

Since the over-densities responsible for the formation of PBHs must be large enough, they will be found at the tail of the probability distribution of density perturbations, which

makes them exponentially rare [9]. At the same time, a large over-density at small scales can modify the large-scale dynamics of the universe in a non-perturbative way. Thus, in order to predict the abundance of PBHs, a precise statistical knowledge of inflationary perturbations is highly desirable.

The hope of the stochastic approach to inflation is that it incorporates quantum corrections to the inflationary dynamics in a non-perturbative way [10]. In this approach, wavelengths that are well outside the horizon are approximated in powers of spatial gradients rather than on amplitudes (as in linear theory). At the same time though, those modes are influenced by the quantum sector by receiving quantum-kicks from stochastic forces generated by the perturbative sub-horizon modes. The success of the stochastic formalism resides in the fact that it allows to reduce a quantum problem into a statistical one and it has been widely used in the literature [10–68].

Since this manuscript is mainly devoted to stochastic inflation, we will briefly review the long history behind it, which will also help us to understand why the paper is organized as it is.

Soon after Starobisnky first proposed the idea of treating the short-wavelength part of the field as a white noise which is included as a source term in the equation of motion for the classical (long-wavelength) field in [10], stochastic inflation immediately gained huge popularity [11–24], manly because of the possibility of solving the Fokker-Planck equation that govern the evolution of the probability distribution of the amplitude of the perturbations of the scalar field during inflation in an exact way for some specific form of the potential, or even for generic potentials with the help of the first time passage analysis [69] as firstly done in [24]. In the references above, the heuristic derivation of Starobinsky, which was only valid in single-field slow-roll inflationary models, was extended beyond SR and multifield inflation.

At the same time, Morikawa et al. put the stochastic formalism on a more firm ground [25,26], where the heuristic derivation of the equations of motion of stochastic inflation done by previous authors was refined. More concretely, they were able to derive the same stochastic equations integrating out the short-wavelength part of the scalar field in the path integral. Although this derivation lead to the same equations of motion as the heuristic argument followed by previous authors, it is very useful in order to understand what are the approximations that lead to the stochastic equations first presented by Starobinsky and its consequences.

The most important of these approximations is probably to compute the variance of the noises in a pure de-Sitter background. This approach has been vastly used in mort applications of the stochastic formalism until today. Examples of this can be found in [27,29–33,35,37–39,50,54,56], but more importantly, with the aid of this approach, the well-known $\delta N$ formalism [70–73] was extended to an stochastic $\delta N$ formalism [36,41,46,47,53,61,62,66] which allow us to compute the probability distribution of the curvature perturbation and hence to connect the results of stochastic inflation with PBH formation as done for example in [52,58,59,63,64,67].

However, computing the noises in a pure de-Sitter background has important consequences, in fact, it was checked in [35,37,40,45,47–49] that, under this approximation, stochastic formalism recovers the standard result of Quantum Field Theory (QFT) for *test* scalar fields on a fixed inflationary background, which give us a hint that thos approach does not correctly take into account the backreaction of the scalar field in the metric. As pointed out in [74], this method misses the coupling between the long- and short-wavelength sectors; more concretely, the background in which the noises should be computed is not exactly de-Sitter, but it is both slow-roll and stochastically corrected. This means that the results obtained with a stochastic formalism that computes the noises in a pure de-Sitter background should not be taken seriously if we are looking for deviations from de-Sitter shuch as in the spectral index. This important limitation of the stochastic formalism as presented by Starobinsky (which will be further studied in Section 6.1.1), led to some authors to present alternative ways of solving the equations of motion of stochastic

inflation beyond the usual Fokker-Plank equation with de-Sitter noises, some examples are the recursive stategy presented in [42–44] or numerical codes as in [67].

Another important simplification that is usually performed in the stochastic framework is to split long- and short-wavelengths modes via a sharp window function (a step function). This choice leads to white noises which are very easy to handle both analytically and numerically. However, it was noted in [28] that this choice of window function leads to some problems in the asymptotic of the noise correlator at large spatial distances. This is why some works with smooth window functions that lead to coloured noises have also been proposed [34,68].

If we now go back to the historical controversies of stochastic inflation, we can find the issue of the time variable in which this formalism must be formulated. Already in one of the first works by Starobinsky [24], it was pointed out that the number of e-folds, (defined as $N = \int H dt$, where $t$ is the cosmic time and $H$ is the Hubble parameter) seem to be a very appropriate time variable. This choice was later justified by connecting stochastic inflation with results form QFT in curved space times [35,37,47]; however, as also pointed out by these references, the choice of $N$ as time variable is only a consequence of writing the stochastic equations in terms only of the scalar field. In fact, one can check that the only way to write all the scalar degrees of freedom in terms solely of the scalar field is to use the uniform-$N$ gauge in the separate universe approach [60], which justifies the use of $N$ as time variable since, by definition, in this gauge it is unperturbed. It was not until recently [65] when the stochastic formalism was formulated in another gauge, making it clear that the choice of $N$ as time variable is only a consequence of the gauge choice.

Finally, it was lately noticed that the stochastic formalism of inflation was leading to some inconsistencies at next-to-leading order in SR parameters even at linear level. For example, it was checked in [55] that the linearization of the stochastic equations do not exactly reproduce the well-known equation of motion of the scalar linear perturbations in the long-wavelength limit (the Mukhanov-Sasaki equation, see Section 4.1). Furthermore, in [57] it was demonstrated that, during an ultra-slow-roll (USR) inflationary regime, some noises that appear in the stochastic equations of motion were incompatible with the rest of the system. The origin of these inconsistencies can be traced back to the use of the separate universe approach in the construction of the stochastic formalism and they can be solved by including the momentum constraint in the separate universe approach [65].

This review will give a detailed derivation of the stochastic formalism to inflation rather than giving a list of results obtained within this formalism. During the derivation we will emphasise all controversial aspects that have been indicated in the above summary of the long history of the stochastic formalism, from the difference between the separate universe approach and the true long-wavelength limit of scalar inhomogeneities to the construction of the noises using an exact de-Sitter background.

The review is organized as follows: after presenting the basic inflationary concepts with the aid of the homogeneous picture of inflation in Section 2, we will start studying inflationary inhomogeneities, focusing on its long-wavelength behaviour. In order to do so we will present different approximations to the exact Arnowitt-Deser-Misner (ADM) equations of Section 3. These approximations are linear perturbation theory (in Section 4), gradient expansion (in Section 5) and the stochastic formalism (in Section 6), which combines the two approximation schemes presented before. Finally, in Section 7 we will give some conclusions.

## 2. Homogeneous Inflationary Scenarios and Slow-Roll Parameters

As we have already mentioned, PBHs represent natural candidates for dark matter (DM) (latest constraints on this idea can be found in [75]). However, one possibility to statistically generate enough PBHs for this to hold one needs, at least, a power spectrum of primordial curvature perturbations several order of magnitudes larger than the one observed in the cosmic microwave background (CMB). Note that this is not the only possibility; one could statistically generate enough PBHs with a large non-gaussianity in

the probability distribution of fluctuations even without enhancing the power spectrum. This scenario is discussed in [76].

It is known that a period of Slow-Roll (SR), of which the predictions of the CMB are based upon, cannot lead to the appropriate power spectrum necessary to generate enough PBHs to match the DM abundance [77] ( For the non-linear relation between the inflationary power spectrum and PBHS abundance, under the assumption of gaussianity, the interested reader can see [78]). Thus, one necessarily needs an inflationary epoch evolving beyond SR. A possibility is the introduction of an inflection point in the inflationary potential [79]. This leads the inflaton to undergo a so-called Ultra-Slow-Roll (USR) phase of inflation [80,81]. Taking into account that the statistics of PBHs from non-gaussian fluctuations has yet to be fully developed. The single field USR option with standard kinetic term seems then to be the best [76]. Such a system is described by the Einstein-Hilbert action with a minimally coupled scalar field i.e.,:

$$ S = \frac{1}{2} \int \sqrt{-g} \left[ M_{PL}^2 R - (\nabla \phi)^2 - 2V(\phi) \right], \tag{1}$$

whose homogeneous solution is an universe described by a Friedman-Lemaitre-Robertson-Walker (FLRW) metric:

$$ ds^2 = -dt^2 + a(t)^2 d\vec{x} \cdot d\vec{x}, \tag{2}$$

where $a(t)$ represents the scale factor.

The equation of motion of the scalar field in the universe described by (2) has the following equation of motion:

$$ \ddot{\phi}^b + 3H^b \dot{\phi}^b + V_{\phi^b}\left( \phi^b \right) = 0, \tag{3}$$

where $V_{\phi^b} \equiv \frac{\partial V(\phi^b)}{\partial \phi^b}$, $H^b \equiv \frac{\dot{a}}{a}$ is the Hubble parameter, and a dot denotes a derivative with respect to the cosmic time $t$. Finally, the super-script "b" stands for "background", the meaning of which will be clarified later on.

The Friedmann equation is

$$ \left( H^b \right)^2 = \frac{1}{3M_{PL}^2} \left( \frac{\left( \dot{\phi}^b \right)^2}{2} + V\left( \phi^b \right) \right). \tag{4}$$

The Slow-Roll (SR) parameters $\epsilon_i$ define the rate of change of the Hubble parameter:

$$ \epsilon_1 \equiv -\frac{\dot{H}^b}{\left( H^b \right)^2} = \frac{\left( \dot{\phi}^b \right)^2}{2H^2 M_{PL}^2} \ll 1; \qquad \epsilon_{i+1} \equiv \frac{\dot{\epsilon}_i}{H\epsilon_i} \quad \text{with} \quad i \geq 1, \tag{5}$$

where, to write the final expressions, we have used the Friedmann equation and the equation of motion of the field.

We can now define different inflationary regimes depending on the values of the $\epsilon_i$s:

- Slow-Roll (SR): The field is slowly rolling down a potential with an almost constant velocity, which makes the acceleration negligible. In this case the equation of motion (3) is approximately

$$ 3H^b \dot{\phi}^b + V_\phi\left( \phi^b \right) \simeq 0. \tag{6}$$

The SR parameters are much smaller than one ($\epsilon_i \ll 1$) and can be written in terms of the potential as

$$ \epsilon_1^{SR} \simeq \frac{1}{2M_{PL}^2} \left( \frac{V_{\phi^b}}{V} \right)^2; \qquad \epsilon_2^{SR} \simeq \frac{2}{M_{PL}^2} \left( \frac{V_{\phi^b \phi^b}}{V} \right) - 4\epsilon_1^{SR}. \tag{7}$$

- Ultra-Slow-Roll (USR): The field is moving along an exactly flat potential $(V_\phi = 0)$, which makes the acceleration relevant. In this case the equation of motion (3) is

$$\ddot{\phi}^b + 3H^b\dot{\phi}^b = 0\,.\tag{8}$$

From (8) one can infer that the velocity of the field (and hence $\epsilon_1$) exponentially decreases, which makes some $\epsilon_i \sim \mathcal{O}(1)$. More precisely:

$$\epsilon_i^{USR} = -6 + 2\epsilon_1^{USR} \qquad \text{when i even.}$$
$$\epsilon_i^{USR} = 2\epsilon_1^{USR} \qquad \text{when i>1 and odd.}\tag{9}$$

As we will show later on, an exponential decrease of $\epsilon_1$ makes the power spectrum of curvature perturbation increase.
- Both SR and USR are, at least approximately, sub-cases of Constant-Roll (CR). Here $\frac{\ddot{\phi}^b}{H^b\dot{\phi}^b} = \kappa$ where $\kappa$ is a constant. SR is realized when $\kappa = 0$ while USR when $\kappa = -3$. We will not analyse further this generic case.

It is important to remark that, given a potential related to PBH formation, the SR and USR phases alternate. Thus, the equations of motion (6) and (8) will always be an approximation of the system.

## 3. Inhomogeneities during Inflation: The ADM Formalism

Although it is very useful in order to understand the inflationary dynamics, the homogeneous picture of inflation presented in Section 2 is not very realistic and some inhomogeneities must be introduced in order to explain the universe we currently observe. Thus, we must define a general metric which is not restricted to homogeneity and isotropy. An option which is very useful in the context of inflation is to work in the so-called Arnowitt-Deser-Misner (ADM) formalism [82]. This framework supposes that the four-dimensional space-time is foliated into a family of three-dimensional hypersurfaces $\Sigma_t$, labeled by its time coordinate $t$.

In order to work in the ADM formalism, it is convenient to write the metric as:

$$ds^2 = g_{\mu\nu}dx^\mu dx^\nu = -\alpha^2 dt^2 + \gamma_{ij}(dx^i + \beta^i dt)(dx^j + \beta^j dt)\,,\tag{10}$$

and the action (1) presented in the previous section becomes:

$$S = \frac{1}{2}\int\sqrt{\gamma}\left[M_{PL}^2\left(\alpha R^{(3)} + \alpha\left(K_{ij}K^{ij} - K^2\right)\right) - 2\alpha V(\phi) + \alpha^{-1}\left(\dot{\phi} - \beta^i\partial_i\phi - \alpha\gamma^{ij}\partial_i\phi\partial_j\phi\right)\right]\,,\tag{11}$$

where we have introduced many new terms:

- $\alpha$ is the lapse function, which measures the rate of flow of proper time with respect to coordinate time $t$ as one moves normally to $\Sigma_t$.
- $\beta^i$ is the shift vector, which measures how much the local spatial coordinate system shifts tangential to $\Sigma_t$, when moving from $\Sigma_t$ to $\Sigma_{t+\delta t}$ along the normal direction to $\Sigma_t$.
- $\gamma_{ij}$ represents the induced metric on the hypersurface $\Sigma_t$, that we we will decompose as $\gamma^{ij} = a^2 e^{2\zeta}\tilde{\gamma}^{ij}$ with det $\tilde{\gamma}^{ij} = 1$ so that the scale factor $a$ is explicitly present.
- $R_{ij}^{(3)}$ is the Ricci tensor of the spatial metric, and hence $R^{(3)} \equiv \gamma^{ij}R_{ij}^{(3)}$.
- Finally, $K_{ij}$ is the extrinsic curvature ($K \equiv \gamma^{ij}K_{ij}$), which is defined as:

$$K_{ij} \equiv -\nabla_i n_j = -\frac{1}{2\alpha}\left(\partial_t\gamma_{ij} - D_i\beta_j - D_j\beta_i\right)\,,\tag{12}$$

where $n_i \equiv (-\alpha, 0, 0, 0)$ is the unit vector normal to the spatial hypersurfaces and $\nabla_\mu$ and $D_i$ are the covariant derivatives with respect to $g_{\mu\nu}$ and $\gamma_{ij}$, respectively.

It is also convenient to decompose the extrinsic curvature into its trace and traceless part as:

$$K_{ij} = \frac{\gamma_{ij}}{3} K + a^2 e^{2\zeta} \tilde{A}_{ij}, \tag{13}$$

where $\tilde{\gamma}^{ij} \tilde{A}_{ij} = 0$. Note that, in the homogeneous limit, i.e., when $\alpha = 1$, $\beta^i = 0$ and $\gamma^{ij} = a^2 \delta_{ij}$ and hence the metric (10) reduces to (2), we can identify the extrinsic curvature with the background Hubble parameter, namely $K = -3\frac{\dot{a}}{a} = -3H^b$. We will then define a more general inhomogeneous Hubble parameter as $H \equiv -\frac{K}{3}$. This makes sense because $K$ represents the expansion rate of the constant time hypersurfaces.

In the ADM formalism, the lapse function and the shift vector act as Lagrange multipliers for the action (11), and hence they generate two constraints: the Hamiltonian and the momentum constraints, which are, respectively:

$$R^{(3)} - \tilde{A}_{ij} \tilde{A}^{ij} + \frac{2}{3} K^2 = \frac{2}{M_{PL}^2} E, \tag{14}$$

$$D^j \tilde{A}_{ij} - \frac{2}{3} D_i K = \frac{1}{M_{PL}^2} J_i, \tag{15}$$

where $E \equiv T_{\mu\nu} n^\mu n^\nu$ and $J_i \equiv T_{\mu j} n^\mu \gamma_i^j$ and $T_{\mu\nu}$ is, in our case, the stress-energy tensor of the scalar field, i.e.,

$$T_{\mu\nu} = \nabla_\mu \phi \nabla_\nu \phi - \frac{1}{2} g_{\mu\nu} (\nabla^\alpha \phi \nabla_\alpha \phi + 2V(\phi)). \tag{16}$$

The two remaining variables, $\gamma_{ij}$ and $K_{ij}$ are the dynamical ones and their evolution equations are given by:

- For $\gamma_{ij}$:

$$(\partial_t - \beta^k \partial_k)\zeta + \frac{\dot{a}}{a} = -\frac{1}{3}(\alpha K - \partial_k \beta^k), \tag{17}$$

$$(\partial_t - \beta^k \partial_k)\tilde{\gamma}_{ij} = -2\alpha \tilde{A}_{ij} + \tilde{\gamma}_{ik}\partial_j \beta^k + \tilde{\gamma}_{jk}\partial_i \beta^k - \frac{2}{3}\tilde{\gamma}_{ij}\partial_k \beta^k. \tag{18}$$

- for $K_{ij}$:

$$(\partial_t - \beta^k \partial_k)K = \alpha \left( \tilde{A}_{ij} \tilde{A}^{ij} + \frac{1}{3} K^2 \right) - D_k D^k \alpha + 4\pi G \alpha (E + S_k^k), \tag{19}$$

$$\begin{aligned}(\partial_t - \beta^k \partial_k)\tilde{A}_{ij} = {}& \frac{e^{-2\zeta}}{a^2} \left[ \alpha \left( R_{ij}^{(3)} - \frac{\gamma_{ij}}{3} R^{(3)} \right) - \left( D_i D_j \alpha - \frac{\gamma_{ij}}{3} D_k D^k \alpha \right) \right] \\ & + \alpha (K\tilde{A}_{ij} - 2\tilde{A}_{ik}\tilde{A}_j^k) + \tilde{A}_{ik}\partial_j \beta^k + \tilde{A}_{jk}\partial_i \beta^k - \frac{2}{3}\tilde{A}_{ij}\partial_k \beta^k \\ & - \frac{8\pi G \alpha e^{-2\zeta}}{a^2} \left( S_{ij} - \frac{\gamma_{ij}}{3} S_k^k \right), \end{aligned} \tag{20}$$

where $S_{ij} = T_{ij}$ and $S_k^k = \gamma^{kl} S_{lk}$.

Finally, and although it can be recovered using the ADM equations just presented, it is also worthy to present here the Klein-Gordon equation for the evolution of the scalar field:

$$\frac{1}{\sqrt{-g}}\partial_\mu [\sqrt{-g} g^{\mu\nu} \partial_\nu \phi] - V_\phi = 0. \tag{21}$$

We will present more detail in the following section, but it is worth remarking here that the homogeneous equations of Section 2 are straightforwardly recovered setting $\alpha = 1$, $\beta^i = 0$ and $\gamma_{ij} = a^2 \delta_{ij}$.

Since the ADM equations are equivalent to the Einstein equations but much easier to implement numerically, they are very useful to study inhomogeneous space-times in an

exact way; however, the numerical methods capable of solving exactly these equations are computationally expensive [83]. Fortunately there exist some very useful approximations that can be done when studying the inflationary universe that, not only allow us to considerably simplify the numerical way of solving the ADM equations, but they even admit some analytical solutions as we will see in the following.

## 4. Linear Perturbation Theory

Since, as explained in the introduction, the deviations from an exactly homogeneous and isotropic FLRW universe that we observe are very small, it makes sense to solve the system of ADM Equations (14)–(21) by expanding them around a FLRW background. If this expansion is done up to first order, we call it linear perturbation theory. There are many reviews on this topic [84–88]; however, since we are taking a slightly different point of view from most of them, we will go through linear perturbation theory in some detail. With this in mind, the ADM metric (10) will be written as:

$$g_{\mu\nu} \simeq g_{\mu\nu}^b + \delta g_{\mu\nu}\,, \tag{22}$$

where $g_{\mu\nu}^b$ is the homogeneous and isotropic metric of Section 2 and $\delta g_{\mu\nu}$ represent the perturbation. This implies that the lapse function, the shift vector and the spatial metric are approximated as:

$$\alpha \simeq 1 + A\,,$$
$$\beta_j \simeq aB_j\,,$$
$$e^{2\zeta} \simeq 1 + 2D\,,$$
$$\tilde{\gamma}_{ij} \simeq \delta_{ij} - 2E_{ij} \tag{23}$$

where $E_{ij}$ is traceless. The reason why $E_{ij}$ is traceless is because is the perturbation of $\tilde{\gamma}_{ij}$, which has unit determinant. Precisely, any matrix with unit determinant can be written as:

$$\tilde{\gamma}_{ij} = e^{-2M_{ij}}\,,$$

where $M_{ij}$ is traceless. Note that the last two linearizations in (23) leads to

$$\gamma_{ij} \simeq a^2 \big[(1 + 2D)\delta_{ij} - 2E_{ij}\big]\,. \tag{24}$$

Finally, the scalar field responsible for inflation must also be linearized, i.e.,

$$\phi \simeq \phi^b + \delta\phi\,. \tag{25}$$

It can be easily shown that, of the linear variables introduced above, $A$, $D$ and $\delta\phi$ transform as scalars under rotations in the background space-time coordinates, $B_i$ as a 3-vector and $E_{ij}$ as a 3D-tensor. This does not mean that the only scalar components are $A$, $D$ and $\delta\phi$. In fact, we know from Euclidean 3D vector calculus that a vector can be decomposed as:

$$B_i = B_i^S + B_i^V \quad \text{with} \quad \partial_i B_j^S - \partial_j B_i^S = 0 \quad \text{and} \quad \partial^i B_i^V = 0\,, \tag{26}$$

and hence

$$B_i^S = \partial_i B\,, \tag{27}$$

where B is some scalar field.

Similarly, for a tensor field we have

$$E_{ij} = E_{ij}^S + E_{ij}^V + E_{ij}^T\,, \tag{28}$$

where

$$E^S_{ij} = \left( \partial_i \partial_j - \frac{1}{3} \delta_{ij} \nabla^2 \right) E \,,$$

$$E^V_{ij} = \frac{1}{2} \left( \partial_j E_i + \partial_i E_j \right) \qquad \text{with} \qquad \partial^i E_i = 0 \,,$$

$$\partial^i E^T_{ij} = 0 \,, \tag{29}$$

$$\delta^{ij} E^T_{ij} = 0 \,, \tag{30}$$

where $E$ is again a scalar field.

The procedure explained above allow us to decompose the perturbations into a scalar, vector and tensor sector. These sectors have the characteristic of evolve independently one from each other at linear order in perturbation theory, which make them easier to handle. During this review we will be mostly focused on scalar perturbations of the metric since they are the ones that couple to the scalar field perturbation $\delta\phi$. This allows us to write the perturbed metric of (23) as:

$$ds^2 = -(1 + 2A)dt^2 + 2a\partial_i B dx^i dt + a^2 \left[ (1 + 2D)\delta_{ij} - 2 \left( \partial_i \partial_j - \frac{1}{3} \delta_{ij} \nabla^2 \right) E \right] dx^i dx^j \,. \tag{31}$$

Similarly to what we have just done with the metric, we can split each one of the ADM equations presented in Section 3 into an homogeneous and a perturbed part as follows:

- Hamiltonian constraint (14).

$$0 = R^{(3)} - \tilde{A}_{ij} \tilde{A}^{ij} + \frac{2}{3} K^2 - \frac{2}{M^2_{PL}} E$$

$$\simeq \left[ 6 \left( H^b \right)^2 - \frac{2}{M^2_{PL}} \left( \frac{\left( \dot\phi^b \right)^2}{2} + V(\phi^b) \right) \right]$$

$$+ \left[ -12 H^b \left( H^b A - \dot{D} + \frac{1}{3} \frac{\nabla^2}{a} B \right) - 4 \frac{\nabla^2}{a^2} \left( D + \frac{1}{3} \nabla^2 E \right) - \frac{2}{M^2_{PL}} \left( \dot\phi^b \dot{\delta\phi} - \left( \dot\phi^b \right)^2 A + V_{\phi^b} \delta\phi \right) \right] \,. \tag{32}$$

Note that in (32) the term inside the first square brackets corresponds to the background Hamiltonian constraint (4) of Section 2, being the term inside the second square brackets the linearization of the Hamiltonian constraint. We will follow this same notation for the rest of the ADM equations.

- Momentum constraint (15).

$$0 = D^j \tilde{A}_{ij} - \frac{2}{3} D_i K - \frac{1}{M^2_{PL}} J_i$$

$$\simeq [0]$$

$$+ \left[ \partial_i \left( -2 H^b A + 2\dot{D} + \frac{2}{3} \nabla^2 \dot{E} + \frac{1}{M^2_{PL}} \dot\phi^b \delta\phi \right) \right] \,. \tag{33}$$

As it can be seen in (33), the momentum constraint do not have a contribution at the background level, which is logical due to the presence of spatial derivatives, which cannot appear in an exactly homogeneous and isotropic global background. However, it does have an homogeneous contribution in the perturbative sector due to the presence of a total spatial derivative, this is of crucial importance for the rest of the review.

- Evolution equation for $\zeta$ (17).

$$
\begin{aligned}
0 = (\partial_t - \beta^k \partial_k)\zeta + \frac{\dot{a}}{a} &= -\frac{1}{3}(\alpha K - \partial_k \beta^k) \\
&\simeq \left[ -H^b + \frac{\dot{a}}{a} \right] \\
&+ \left[ \dot{D} - H^b A - \delta H - \frac{1}{3}\frac{\nabla^2}{a}B \right],
\end{aligned}
\tag{34}
$$

where we have used the identification $K \equiv -3H$ explained below Equation (13). From (34) we can easily identify the perturbation of the Hubble parameter, i.e.,

$$
H = H^b + \delta H = H^b + \left[ -H^b A + \dot{D} - \frac{1}{3}\frac{\nabla^2}{a}B \right].
\tag{35}
$$

- Evolution equation for $\tilde{A}_{ij}$ (18).

$$
\begin{aligned}
0 = (\partial_t - \beta^k \partial_k)\tilde{\gamma}_{ij} &= -2\alpha \tilde{A}_{ij} + \tilde{\gamma}_{ik}\partial_j \beta^k + \tilde{\gamma}_{jk}\partial_i \beta^k - \frac{2}{3}\tilde{\gamma}_{ij}\partial_k \beta^k \\
&\simeq [0] \\
&+ \left[ 2\tilde{A}_{ij} - 2\left( \partial_i \partial_j - \frac{1}{3}\delta_{ij}\nabla^2 \right)\left( \frac{B}{a} + \dot{E} \right) \right].
\end{aligned}
\tag{36}
$$

- Evolution equation for $K$ (19).

$$
\begin{aligned}
0 = (\partial_t - \beta^k \partial_k)K &= \alpha\left( \tilde{A}_{ij}\tilde{A}^{ij} + \frac{1}{3}K^2 \right) - D_k D^k \alpha + 4\pi G \alpha (E + S_k^k) \\
&\simeq \left[ -3\dot{H}^b - 3\left( H^b \right)^2 - \frac{1}{M_{PL}^2}\left( \left( \dot{\phi}^b \right)^2 - V(\phi^b) \right) \right] \\
&+ \left[ 6\dot{H}^b A + 3H^b \dot{A} - 3\ddot{D} + \frac{\nabla^2}{a}\dot{B} + \frac{\nabla^2}{a^2}A + 6\left( H^b \right)^2 A - 6H^b \dot{D} + H^b \frac{\nabla^2}{a}B \right. \\
&\left. - \frac{1}{M_{PL}^2}\left( 2\dot{\phi}^b \dot{\delta\phi} - 2\left( \dot{\phi}^b \right)^2 A - V_{\phi^b}\delta\phi \right) \right].
\end{aligned}
\tag{37}
$$

- Evolution equation for $\tilde{A}_{ij}$ (20).

$$
\begin{aligned}
0 = (\partial_t - \beta^k \partial_k)\tilde{A}_{ij} &- \frac{e^{-2\zeta}}{a^2}\left[ \alpha\left( R_{ij}^{(3)} - \frac{\gamma_{ij}}{3}R^{(3)} \right) - \left( D_i D_j \alpha - \frac{\gamma_{ij}}{3}D_k D^k \alpha \right) \right] \\
&- \alpha(K\tilde{A}_{ij} - 2\tilde{A}_{ik}\tilde{A}_j^k) - \tilde{A}_{ik}\partial_j \beta^k - \tilde{A}_{jk}\partial_i \beta^k + \frac{2}{3}\tilde{A}_{ij}\partial_k \beta^k + \frac{8\pi G \alpha e^{-2\zeta}}{a^2}\left( S_{ij} - \frac{\gamma_{ij}}{3}S_k^k \right) \\
&\simeq [0] \\
&+ \left[ \left( \partial_i \partial_j - \frac{1}{3}\delta_{ij}\nabla^2 \right)\left( \frac{\dot{B}}{a} + 2\frac{H^b B}{a} + \ddot{E} + 3H^b \dot{E} + A + D + \frac{1}{3}\nabla^2 E \right) \right].
\end{aligned}
\tag{38}
$$

- Evolution equation for the scalar field (21).

$$
\begin{aligned}
0 &= \frac{1}{\sqrt{-g}}\partial_\mu\left[ \sqrt{-g}g^{\mu\nu}\partial_\nu \phi \right] - V_\phi \\
&\simeq \left[ \ddot{\phi}^b + 3H^b \dot{\phi}^b + V_{\phi^b}(\phi^b) \right] \\
&+ \left[ \ddot{\delta\phi} + 3H^b \dot{\delta\phi} + V_{\phi^b \phi^b}\delta\phi - \frac{\nabla^2}{a^2}\delta\phi + 2V_{\phi^b}A - \dot{\phi}^b\left( \dot{A} - 3\dot{D} + \frac{\nabla^2}{a}B \right) \right].
\end{aligned}
\tag{39}
$$

Once we have seen how linear perturbation theory works and what are the linear equations that describe small inhomogeneities during an inflationary epoch, it is important to have a physical intuition about what a perturbation of the metric really means. By definition, a perturbation is the difference between the value of a quantity in the real and inhomogeneous space-time and its value on the idealized FLRW background. This seems trivial; however, in order to make such a comparison, it is necessary to compute these two values at the same space-time point. Since the quantities to compare live in different space-times, we require a pointwise correspondence between them, which is given by a coordinate system $x^\mu$ such that the point $P^b$ in the background space-time and the point P in the perturbed space-time, which have the same coordinate values, correspond to each other.

The freedom in the choice among these coordinate systems is called the gauge choice. The way the different gauges are related in linear perturbation theory (for gauge transformations beyond linear perturbation theory see for example [89]) is via an infinitesimal gauge transformation of the coordinates:

$$\tilde{x}^\mu = x^\mu + \delta x^\mu \tag{40}$$

We can split the vector $\delta x^\mu$ into its time an space components $\delta x^\mu = (\lambda^0, \lambda^i)$, and following the same idea as when we decomposed the perturbations in the metric, $\lambda^i$ can be written as $\lambda^i = \lambda^i_\perp + \partial^i \eta$, where $\lambda^i_\perp$ is a 3-dimensional divergenless vector and $\eta$ is a scalar function. In terms of these functions, we can impose the gauge invariance of the perturbed metric (31) to deduce the gauge transformation of each one of the scalar variables in the metric:

$$D \to \tilde{D} = D + aH^b \lambda^0 + \frac{1}{3}\nabla^2 \eta \,,$$
$$A \to \tilde{A} = A + aH^b \lambda^0 + a\dot{\lambda}^0 \,,$$
$$E \to \tilde{E} = E - \eta \,,$$
$$B \to \tilde{B} = B + a\dot{\eta} - \lambda^0 \,. \tag{41}$$

Finally, the scalar field perturbation will transform as:

$$\delta\phi \to \tilde{\delta\phi} + a\dot{\phi}\lambda^0 \,. \tag{42}$$

From (41) and (42) we can clearly see that the freedom on the choice of the gauge allow us to set two out of the five scalar perturbations to zero by choosing $\eta$ and $\lambda^0$ accordingly. This reduces the scalar degrees of freedom to three (which further reduces to two when using the ADM equations for a single scalar field), which can be written in terms of gauge invariant, and hence physical, variables: the two Bardeen potentials [90]

$$\Psi \equiv -D - \frac{1}{3}\nabla^2 E - aH^b\left(B + a\dot{E}\right),$$
$$\Phi \equiv A + aH^b\left(B + a\dot{E}\right) + a\frac{d}{dt}\left(B + a\dot{E}\right), \tag{43}$$

and the Mukhanov-Sasaki (MS) variable

$$Q \equiv \delta\phi - \frac{\dot{\phi}^b}{H^b}\left(D + \frac{1}{3}\nabla^2 E\right). \tag{44}$$

During this review we will pay special attention to the MS variable. In fact, it can be shown that, by rearranging the linearized ADM equations of (32)–(39), the MS variable

follows a simple equation of motion which, written in terms of the SR parameters defined in Section 2, is:

$$\ddot{Q} + 3H^b\dot{Q} + \left[-\frac{\nabla^2}{a^2} + \left(H^b\right)^2\left(-\frac{3}{2}\epsilon_2 + \frac{1}{2}\epsilon_1\epsilon_2 - \frac{1}{4}\epsilon_2^2 - \frac{1}{2}\epsilon_2\epsilon_3\right)\right]Q = 0. \tag{45}$$

In order to solve (45), we must take into account that subhorizon scales during inflation are microscopic, and hence we must study the behavior of the inflationary scalar field using quantum field theory (QFT). The way to proceed is similar to what we do when quantizing the harmonic oscillator: $\hat{Q}(\mathbf{x}, t)$, which is now a quantum operator, can be expressed in Fourier space as:

$$\hat{Q}(\mathbf{x}, t) = \int \frac{d\mathbf{k}}{(2\pi)^{3/2}} \hat{Q}_{\mathbf{k}}(t), \tag{46}$$

where

$$\hat{Q}_{\mathbf{k}}(t) = e^{-i\mathbf{k}\cdot\mathbf{x}}Q_{\mathbf{k}}(t)a_{\mathbf{k}} + e^{-i\mathbf{k}\cdot\mathbf{x}}Q_{\mathbf{k}}^*(t)a_{\mathbf{k}}^\dagger, \tag{47}$$

and $a_{\mathbf{k}}$ and $a_{\mathbf{k}}^\dagger$ are the usual creation and annihilation operators, that satisfy the usual commutation relation:

$$[a_{\mathbf{k}}, a_{\mathbf{k}'}^\dagger] = \delta^{(3)}(\mathbf{k} - \mathbf{k}') \tag{48}$$

With this construction, the variable $Q_{\mathbf{k}}$ of (47) is the solution of the MS Equation (45) in Fourier space, i.e.,:

$$\ddot{Q}_{\mathbf{k}} + 3H^b\dot{Q}_{\mathbf{k}} + \left[\frac{k^2}{a^2} + \left(H^b\right)^2\left(-\frac{3}{2}\epsilon_2 + \frac{1}{2}\epsilon_1\epsilon_2 - \frac{1}{4}\epsilon_2^2 - \frac{1}{2}\epsilon_2\epsilon_3\right)\right]Q_{\mathbf{k}} = 0 \tag{49}$$

If we impose the Bunch-Davies vacuum [91] to be recovered at early time, we can only write analitically a solution for (49) if a specific condition is satisfied. In order to see what is this condition we will define the variable $z$ as $z = a\frac{\dot{\phi}^b}{H^b}$ and the conformal time $\tau$ as $a d\tau = dt$, with this, the analytical solution of (49) in terms of the conformal time only exists if

$$\nu^2 \equiv \frac{1}{4} + \tau^2\frac{1}{z}\frac{d^2z}{d\tau^2} \tag{50}$$

is a constant. It is easy to check that this is the case generically; in fact we can integrate by parts $\tau = \int \frac{dt}{a}$ to get:

$$\tau \simeq -\frac{1}{aH^b}(1 + \mathcal{O}(\epsilon_1)).$$

See for example Appendix E of [65] for the derivation. The equation above, together with the fact that

$$\frac{1}{z}\frac{d^2z}{d\tau^2} = a^2\left(H^b\right)^2\left(2 - \epsilon_1 + \frac{3}{2}\epsilon_2 - \frac{1}{2}\epsilon_1\epsilon_2 + \frac{1}{4}\epsilon_2^2 + \frac{1}{2}\epsilon_2\epsilon_3\right),$$

makes $\nu$ defined in (50) to be a constant generically up to $\mathcal{O}(\epsilon_1)$. Note that, since in SR, $\epsilon_i$ is also a constant up to $\mathcal{O}(\epsilon_1)$, in this case we have that $\nu^{SR}$ is constant up to $\mathcal{O}\left(\left(\epsilon_i^{SR}\right)^2\right)$. The analytical solution in this case is:

$$Q_{\mathbf{k}} = e^{\frac{i}{2}\pi\left(\nu+\frac{1}{2}\right)}\frac{\sqrt{\pi}}{2a}\sqrt{-\tau}H_\nu^{(1)}(-k\tau), \tag{51}$$

where $H_\nu^{(1)}$ is the Hankel function of first class.

In order to finish this reminder of linear perturbation theory, let us introduce a very useful quantity that characterizes the properties of the perturbations: the power spectrum. We will give a physical interpretation of the power spectrum later on and for the moment

we will focus only on its mathematical definition. The power spectrum of a generic quantity $X(\mathbf{x}, t)$ is defined as the Fourier transform of the two-point correlation function, i.e.:

$$\langle 0 | X(\mathbf{x}_1, t) X(\mathbf{x}_2, t) | 0 \rangle = \int \frac{d\mathbf{k}}{(2\pi)^3} |X_{\mathbf{k}}(t)|^2 \equiv \int \frac{dk}{k} \mathcal{P}_X(k, t) \frac{\sin(kr)}{kr} \, , \tag{52}$$

where $r = |\mathbf{x}_1 - \mathbf{x}_2|$ and $k = |\mathbf{k}|$. From (52) it is clear that the power spectrum is defined as:

$$\mathcal{P}_X(k, t) \equiv \frac{k^3}{2\pi^2} |X_{\mathbf{k}}|^2 \, . \tag{53}$$

Another interesting quantity derived from the power spectrum is the spectral index $n_s^X - 1$, which is defined as:

$$n_s^X - 1 \equiv \frac{d \log \mathcal{P}_X}{d \log k} \, . \tag{54}$$

### 4.1. Long Wavelength Limit of Linear Perturbation Theory

As explained in the introduction, many quantities of interest such as the power spectrum of the scalar tilt are usually computed in the long wavelength limit, or superhorizon scales. This makes sense because, as long as inflation is taking place, the exponential expansion of the universe stretch the perturbations of the quantum fields from microscopic to cosmological scales. In this way, the inhomogeneities that re-enter the horizon once inflation has finished and hence the ones of interest today, were in its long-wavelength limit when inflation ended. In this section we will take a close look to this limit and its physical consequences. We will study the behaviour of perturbations with characteristic wavelength $L$ much larger that a local Hubble radius $H_l^{(-1)}$ in such a way that if we consider $L$ to be infinitely large compared with $H_l^{(-1)}$, we can interpret the region inside $H_l^{(-1)}$ as a local universe without perturbations, i.e., homogeneous and isotropic. In other words, if $\frac{L}{H_l^{(-1)}} \to \infty$, or equivalently, if $\frac{k}{a_l H_l} \simeq k\tau_l \to 0$, then the region inside $H_l^{(-1)}$ represents a local FLRW universe. With this in mind, let us study the long wavelength limit of perturbations during inflation. In order to do so we will take two different point of view: the first one is very intuitive and we will call it the $k \to 0$ limit, the second one is slightly less intuitive but it is very useful when dealing with non-linear perturbations, this is the so-called linear separate universe approach of $k = 0$ case. Although they are very similar, the two point of view are not exactly equivalent as we will see in the following.

#### 4.1.1. $K \to 0$ Limit

Since each spatial derivative introduces a factor $k$ in Fourier space, it would be logical to think that in the $k \to 0$ limit of the ADM equations we must neglect any term that contains a spatial derivative (This will be done in the next subsection). However, by doing that, we would be neglecting terms that actually contribute in the $k \to 0$ limit.

In the following we will enumerate the two cases in which neglecting a term with a spatial derivative would be too naive:

1. Non-local terms.
   As an example, let us imagine a re-scaling of the spatial coordinates

$$x^i \to \tilde{x}^i = (1 + \delta) x^i \, . \tag{55}$$

It is easy to realize that if we rewrite $\delta$ in terms of the transformations parameters of (41) we get:

$$\nabla^2 \eta = \delta \, . \tag{56}$$

The fact that a transformation as the one described in (55) is perfectly allowed in a FLRW universe together with (56) and the transformations rules of the perturbed

metric variables of (41) immediately tell us that variables like $\nabla^2 E$ cannot generically be neglected in the long wavelength limit even if they contain a Laplacian. In order to avoid this problem, we will only neglect terms that contain extra spatial derivatives, by extra we mean that we will neglect a term like $\partial_i X$ if and only if it is compared with $X$, but we will not neglect it if it appear alone.

2. Equations with overall spatial derivatives.

A clear example is the momentum constraint of (33). Because it contains a total spatial derivative, it gives non-trivial information even in the $k \to 0$ limit, namely:

$$H^b A - \dot{D} - \frac{1}{3}\nabla^2 \dot{E} - \frac{1}{2M_{PL}^2}\dot{\phi}^b \delta\phi = 0 \,. \tag{57}$$

The same happens with the evolution equation for $\tilde{A}_{ij}$ (38).

Taking these two important point into account one can re-derive the equation of motion for the MS variable $Q$ in the $k \to 0$ limit. The result is, as expected, the Equation (45) but without the Laplacian, i.e.,

$$\ddot{Q} + 3H^b \dot{Q} + \left(H^b\right)^2 \left(-\frac{3}{2}\epsilon_2 + \frac{1}{2}\epsilon_1\epsilon_2 - \frac{1}{4}\epsilon_2^2 - \frac{1}{2}\epsilon_2\epsilon_3\right)Q = 0 \,. \tag{58}$$

If we want to know the solution for Equation (58) we can simply take the $k\tau \to 0$ limit of the solution (51) obtained before (we will do that later on); however, this would be restricted to $\nu$ (defined in (50)) being a constant. In this section we will solve independently the long wavelength equation of motion for $Q$ without any assumption, i.e., valid at all orders in slow-roll parameters. The price to pay is that, since the initial conditions are given at sub-horizon scales and (58) is only valid at super-horizon scales, we will not specify any initial condition here.

It can be checked that (58) can be written as a total derivative as follows:

$$\frac{2M_{PL}^2 H^b}{a^3 \dot{\phi}^b} \frac{d}{dt}\left[\frac{a^3 \dot{\phi}^b}{2M_{PL}^2}\left(\frac{\dot{Q}}{H^b} - \frac{\epsilon_2}{2}Q\right)\right] = 0 \,. \tag{59}$$

For convenience, we will define the comoving curvature perturbation as:

$$\mathcal{R}_c \equiv -\frac{H^b}{\dot{\phi}^b}Q \,. \tag{60}$$

It terms of $\mathcal{R}_c$, (59) takes a very simple form:

$$a^3 \epsilon_1 \dot{\mathcal{R}}_c = C_1 \,, \tag{61}$$

where $C_1$ is a constant. Therefore, the solution of the MS equation in the long wavelength limit (58) is

$$Q = \frac{\dot{\phi}^b}{H^b}\mathcal{R}_c = C_2\frac{\dot{\phi}^b}{H^b} + C_1\frac{\dot{\phi}^b}{H^b}\int \frac{dt}{a^3\epsilon_1} \,, \tag{62}$$

where $C_2$ is also a constant. Solution (62) is the well-known exact solution in the $k \to 0$ limit for the single component scalar field case [85,92]. The term proportional to $C_1$ is usually known as decaying mode, name inherited from its SR behaviour where $\dot{\phi}^b$ is roughly constant, since $a \sim e^{-H^b t}$ during inflation, we can clearly see that the term proportional to $C_1$ decays as $e^{-3H^b t}$ during SR. However, this is not the case beyond SR, for example in USR we have $\dot{\phi}^b \sim e^{-3H^b t}$ (see (8)) and hence $\epsilon_1 \sim e^{-6H^b t}$, which makes the term proportional to $C_1$ be approximately constant whereas the one proportional to $C_2$ decays, contrary to what happens in SR.

4.1.2. $K = 0$ or Linear Separate Universe Approach

The second point of view to solve linear perturbation theory in the long wavelength limit is the linear separate universe approach [93–95], in this case we will forget for a moment about the perturbed ADM equations and we will take advantage of the fact that, in the long wavelength limit, each local patch represents an homogeneous and isotropic universe such that its line element is:

$$ds_l^2 = -dt_l^2 + a_l(t_l)^2 \delta_{ij} dx_l^i dx_l^j, \tag{63}$$

where, as before, the subscript $l$ stands for local. As a consequence, the constraints and equations of motion will be the same as the ones presented in Section 2, i.e.,

$$3M_{PL}^2 H_l^2 = \frac{1}{2}\left(\frac{d}{dt_l}\phi_l\right)^2 + V(\phi_l),$$

$$\frac{d}{dt_l}H_l + H_l^2 = -\frac{1}{3M_{PL}^2}\left[\left(\frac{d}{dt_l}\phi_l\right)^2 - V(\phi_l)\right],$$

$$\frac{d^2\phi^l}{dt_l^2} + 3H_l\frac{d}{dt_l}\phi_l + V_{\phi_l}(\phi_l) = 0. \tag{64}$$

The first thing to realize here is that the momentum constraint is missing. This is because in the separate universe approach we are not only assuming that each patch is homogeneous and isotropic (we were also making this assumption in the $k \to 0$ limit where the momentum constraint plays a role), but we are also assuming that each patch evolve independently from each other. Since the momentum constraint gives information about the interaction between the different FLRW patches due to the presence of a spatial derivative, it makes sense that it does not appear in the linear separate universe approach. Before proceeding, let us clarify here that when we talk about the information about the interaction between different patches encoded in the momentum constraint, which makes these patches evolve in a not completely independent way, we do not claim that what happens in one patch will affect the others, but rather that all the patches as an ensemble must satisfy some conditions (given by the momentum constraint) that are absent if we look to a single patch but that are important when comparing them.

Within this approximation, we can only see the effect of perturbations if we compare different patches between them or with the global background. Knowing that the differences must be perturbative, we can follow our results from linear perturbation theory and write:

$$H_l \simeq H^b + \delta H = H^b - H^b A + \dot{D} - \frac{1}{3}\frac{\nabla^2}{a}B,$$

$$dt_l \simeq (1 + A)dt,$$

$$\phi_l \simeq \phi^b + \delta\phi. \tag{65}$$

The term $\frac{1}{3}\frac{\nabla^2}{a}B$ can be set to zero without loss of generality, this is because since each patch is independent from each other, we can always choose an orthogonal threading for all of them, in which $\beta_i$ (and hence $B$) is zero. Another way of reasoning is that, as checked in the $k \to 0$ limit of perturbation theory, a term with a Laplacian can only be important in the long wavelength limit if it contains non-local information, but at the same time, non-local terms would give some information about the interaction between local patches, which is in contradiction with the absence of the momentum constraint (and hence with the separate universe assumption) as explained above.

If we also take into account that the scalar perturbation related with the traceless part of the spatial metric (i.e., $E$) does not appear in (65) so neither in (64), we can set

$$\nabla^2 B^{sep} = \nabla^2 E^{sep} = 0. \tag{66}$$

in the separate universe approach (as indicated with the superscript "sep"). Note that, although (66) coincides with the Newtonian (or longitudinal) gauge, this is not a gauge choice, but a consequence of the separate universe approach.

With this in mind, we can write the equations in (64) in terms of the global background and perturbations over it as follows:

$$-6\left(H^b\right)^2 A^{sep} + 6H^b \dot{D}^{sep} = \frac{1}{M_{PL}^2}\left(\dot{\phi}^b \delta\phi^{sep} - \left(\dot{\phi}^b\right)^2 A^{sep} + V_{\phi^b}\delta\phi^{sep}\right),$$

$$6\dot{H}^b A^{sep} + 3H^b \dot{A}^{sep} - 3\ddot{D}^{sep} + 6\left(H^b\right)^2 A^{sep} - 6H^b \dot{D}^{sep} = \frac{1}{M_{PL}^2}\left(2\dot{\phi}^b \dot{\delta\phi}^{sep} - \left(\dot{\phi}^b\right)^2 A^{sep} - V_{\phi^b}\delta\phi^{sep}\right),$$

$$\ddot{\delta\phi}^{sep} + 3H^b \dot{\delta\phi}^{sep} + V_{\phi^b\phi^b}\delta\phi^{sep} = -2V_{\phi^b}A^{sep} + \dot{\phi}^b\left[\dot{A}^{sep} - 3\dot{D}^{sep}\right]. \tag{67}$$

As one can see from (67), the linear separate universe approach does not use the whole system of ADM equations presented in Section 4, it only uses the long-wavelength version of: (a) the Hamiltonian constraint (32), (b) the evolution equation for the trace of the extrinsic curvature (37) and (c) the equation of motion of the scalar field (39). All of them setting both $\nabla^2 B^{sep}$ and $\nabla^2 E^{sep}$ to zero accordingly with the separate universe assumption.

In order to see what are the consequences of this reduction in the number of equations and variables when comparing with the $k \to 0$ limit of Section 4.1.1, we can follow a similar procedure to what we did when defining the MS variable and write down a "separate-universe" gauge invariant variable [96,97], meaning that it is gauge invariant under time reparametrizations.

$$Q^{sep} \equiv \delta\phi^{sep} - \frac{\dot{\phi}^b}{H^b}D^{sep}. \tag{68}$$

Similarly to what we did with the MS Equation (45), we can rearrange the equations of the linear separate universe approach (67) and write a single equation of motion for $Q^{sep}$, the result, in terms of the SR parameters, is:

$$\ddot{Q}^{sep} + 3H^b\left(1 + \frac{\epsilon_1\epsilon_2}{3(3-\epsilon_1)}\right)\dot{Q}^{sep} + \left(H^b\right)^2\left(-\frac{3}{2}\epsilon_2 + \frac{1}{2}\epsilon_1\epsilon_2 - \frac{1}{4}\epsilon_2^2 - \frac{1}{2}\epsilon_2\epsilon_3 - \frac{\epsilon_1\epsilon_2^2}{2(3-\epsilon_1)}\right)Q^{sep} = 0. \tag{69}$$

Comparing (69) with (58) we can now clearly identify two extra terms that appear when assuming that each local patch evolves independently from each other, i.e., when using the separate universe approach. These terms are $\mathcal{O}(\epsilon_1\epsilon_2)$ [55], being strongly dependent on the inflationary regime, for example, they are $\mathcal{O}(\epsilon_1^{USR})$ in USR (where $\epsilon_2 \simeq -6$) and $\mathcal{O}\left(\left(\epsilon_1^{SR}\right)^2\right)$ in SR.

In order to better quantify this difference we will solve (69) in a similar way as done with (58) and compare the results. We can again write (69) as a total derivative as follows

$$\frac{2V}{3a^3 H^b \dot{\phi}^b}\frac{d}{dt}\left[\frac{3a^3\left(H^b\right)^2\dot{\phi}^b}{2V}\left(\frac{\dot{Q}^{sep}}{H^b} - \frac{\epsilon_2}{2}Q^{sep}\right)\right] = 0. \tag{70}$$

Defining a "separate universe" comoving curvature perturbation as

$$\mathcal{R}_c^{sep} \equiv -\frac{H^b}{\dot{\phi}^b}Q^{sep}, \tag{71}$$

we can write (70) as

$$\frac{3a^3\left(\dot{\phi}^b\right)^2}{2V} = C_1', \tag{72}$$

where $C_1'$ is a constant. Hence, the solution for $Q^{sep}$ is:

$$Q^{sep} = C_2' \frac{\dot{\phi}^b}{H^b} + C_1' \frac{\dot{\phi}^b}{H^b} \int \left( \frac{1}{a^3 \epsilon_1} - \frac{1}{3a^3} \right).$$

(73)

We can see that the solutions for $Q^{sep}$ of (73) and for $Q$ of (62) differ by an extra term in the decaying mode, which is obviously due to the difference of $\mathcal{O}(\epsilon_1 \epsilon_2)$ in the equation of motion. The importance of this extra term depends on the inflationary regime, being more important beyond SR, where the term proportional to $C_1'$ does not decay (see discussion below (62)).

Since the main difference between the linear separate universe approach and the $k \to 0$ limit is the inclusion of the non-trivial information of the momentum constraint at large scales, which is not taken into account in the former, one could think that the two solutions will coincide if we impose the solution (73) to satisfy the momentum constraint; however, this is not the case. In fact, in the following we are going to check that the mode proportional to $C_1'$, which contains the new extra term, does not represent the $k \to 0$ limit of some solution to the perturbations equations with $k \neq 0$. This should not be a surprise since the correct solution for the MS variable in the long wavelength limit is given by (62). In order to check that, let us rearrange equations (65) to write

$$H^b A^{sep} - \dot{D}^{sep} - \frac{\dot{\phi}^b}{2M_{PL}^2} \delta\phi^{sep} = -\frac{H^b \dot{\phi}^b}{2V} \left( \frac{\dot{Q}^{sep}}{H^b} - \frac{\epsilon_2}{2} Q^{sep} \right).$$

(74)

If we want solution (73) to be the $k \to 0$ solution to the perturbation equations with $k \neq 0$ we must be sure that our solution satisfies the momentum constraint (although it is lost in the separate universe approach.) because, as we know, this constraint gives non-trivial information even at large scales. What would be the momentum constraint in the linear separate universe approach is

$$H^b A^{sep} - \dot{D}^{sep} - \frac{\dot{\phi}^b}{2M_{PL}^2} \delta\phi^{sep} = 0,$$

(75)

which, comparing with (74), immediately implies:

$$\frac{\dot{Q}^{sep}}{H^b} - \frac{\epsilon_2}{2} Q^{sep} = 0.$$

(76)

From condition (76) and the equation of motion (70) we can see that, if the momentum constraint of (75) is satisfied at initial time, it will be always satisfied so one could think that the problem of ignoring the momentum constraint in the linear separate universe approach is simply solved by an appropriate choice of initial conditions; however, the only mode that satisfies (76) is the mode proportional $C_2'$ in the solution (73), which means that the mode proportional to $C_1'$ therein does not represent a $k \to 0$ solution of the perturbation equations with $k \neq 0$ and it must be set to zero. However, setting $C_1' = 0$ means that we are losing the term proportional to $C_1$ in the correct solution of (62), which can be important beyond SR. Thus, imposing that the momentum constraint must be satisfied in the separate universe approach does not give the correct solution for the long-wavelength limit of the MS variable and hence we have to use the whole set of ADM equations in the long wavelength limit to describe the correct dynamics (at all orders in SR parameters) of the MS variable at super-horizon scales. Let us indicate here that although there exist some ways to recover the correct $k \to 0$ limit of the MS variable using only the linear separate universe approach (see [96,97]), we will not study them here because they are restricted to linear perturbation theory and, as we will see later on, we are planning to use the long wavelength limit of inhomogeneities beyond perturbation theory.

For clarity purposes, let us summarize the findings of this section until now:

- Although both the $k \to 0$ and the linear separate universe approach consider local homogeneous and isotropic universes, the $k \to 0$ limit allow some interaction between them whereas the linear separate universe approach assume that they evolve independently. Mathematically speaking, the linear separate universe approach ignore possible non-local terms and equations with overall spatial derivatives, both present in the $k \to 0$ limit.
- As a consequence, the equation of motion for $Q^{sep}$ differ by terms of $\mathcal{O}(\epsilon_1 \epsilon_2)$ from the equation of motion for $Q$ in the long wavelength limit. This difference induces an extra decaying term, namely:

$$Q^{sep} = Q(k \to 0) + C_1' \frac{\dot{\phi}^b}{H^b} \int \frac{1}{3a^3} dt \tag{77}$$

- The difference (77) always decays so one could think that it can be safely ignored; however, its importance strongly depends on the inflationary regime. For example, in USR we have $Q^{sep} - Q(k \to 0) \sim \mathcal{O}(\epsilon_1)$ so if we want to be precise enough when studying the long-wavelength limit of perturbation theory, we should use the $k \to 0$ limit rather than the linear separate universe approach.
- Finally, imposing the momentum constraint to be satisfied in the separate universe approach not only does not solve the difference between $Q^{sep}$ and $Q(k \to 0)$, but it makes it worse.

### 4.2. Linear Perturbation Theory and Pbhs

In order to finish this section we will study the behaviour of the power spectrum of the inflationary regimes that can lead to the formation of PBHs in the long wavelength limit (The reader interested in PBHs as probes for the physics of the very early universe in a more detailed way can read the following reviews: [98,99]). We will do that in a very qualitative way and at leading order in SR parameters, this is why we will simply use the $k\tau \to 0$ limit of solution (51), where the initial conditions are properly specified. If we expand the Hankel function of first order we have:

$$\lim_{k\tau \to 0} Q_{\mathbf{k}} = \lim_{k\tau \to 0} e^{\frac{i}{2}\left(\nu + \frac{1}{2}\right)} \frac{\sqrt{\pi}}{2a} \sqrt{-\tau} H_\nu^{(1)}(-k\tau) \simeq -i e^{\frac{i}{2}\left(\nu + \frac{1}{2}\right)} \frac{2^{\nu-1}}{a\sqrt{\pi}} \sqrt{-\tau}(-k\tau)^{-\nu} \Gamma[\nu], \tag{78}$$

where $\Gamma[\nu]$ is the Euler gamma.

For the formation of PBH we are interested in the power spectrum of the comoving curvature perturbation, i.e.,

$$\mathcal{P}_{\mathcal{R}_c} = \frac{k^3}{2\pi^2} \left(\frac{H^b}{\dot{\phi}^b}\right)^2 |Q_{\mathbf{k}}|^2 \simeq \frac{\left(H^b\right)^4}{\pi^3 (\dot{\phi}^b)^2} \left(\frac{k}{2}\right)^{3-2\nu} \left(\frac{1}{aH^b}\right)^{3-2\nu} \Gamma[\nu]^2, \tag{79}$$

where we have used $\tau \simeq -\frac{1}{aH^b}$.

For USR and SR we have $\nu \simeq \frac{3}{2}$ and hence the power spectrum is roughly scale invariant:

$$\mathcal{P}_{\mathcal{R}_c}^{SR} \simeq \mathcal{P}_{\mathcal{R}_c}^{USR} \simeq \frac{\left(H^b\right)^4}{4\pi^2 (\dot{\phi}^b)^2} = \frac{\left(H^b\right)^2}{8\pi^2 M_{PL}^2 \epsilon_1}. \tag{80}$$

However, and although the k dependence of the comoving curvature power spectrum for USR and SR is roughly the same, and hence $n_s^{\mathcal{R}_c} - 1 \sim \mathcal{O}(\epsilon_1)$ for both, this is not true for the time dependence. In fact we have

$$\mathcal{P}_{\mathcal{R}_c}^{SR} \sim \text{constant}, \qquad \mathcal{P}_{\mathcal{R}_c}^{USR} \sim e^{6H^b t}. \tag{81}$$

An exponential growth of the power spectrum at super-horizon scales as it happens in USR has important consequences for the formation of PBH. In fact, by definition of the power spectrum, $\mathcal{P}_{\mathcal{R}_c}$ give us the variance of the probability distribution that follow the amplitude of the perturbations $\left(\mathcal{R}_{\mathbf{k}_1}\right)_c$ with characteristic wavenumber $\mathbf{k}_1$, therefore, a growth in the power spectrum can be interpreted as a growth in the variance or, equivalently, a spreading of the probability distribution. This means that high values for the amplitude of the perturbations placed at the tail of the probability distribution are much more probables if the power spectrum grows. Finally, and since the high values for the amplitude of the curvature perturbations are the ones that collapse to form a PBH when they re-enter the horizon, we can conclude that inflationary regimes beyond SR favor the creation of PBH.

The argument given above is very hand-waving but it is enough to remark the importance of inflationary regimes beyond SR when talking about PBH. However, if we want to make actual predictions about the mass and abundances of PBH we need to study inflationary dynamics in a much more precise way, in fact, the tail of the probability distribution for the amplitude of perturbations, which is where the perturbations responsible for the formation of PBH are located, is very sensitive to any small change in the power spectrum (or in the higher order correlators as the bispectrum). This is the reason why a description of the perturbations at all orders in SR is highly desirable, making the linear separate universe approach not the best approximation to study the tail of the probability distribution of the amplitude of perturbations generated during a inflationary regime beyond SR, i.e., to study the formation of PBH. The reason is that, generically, $Q^{sep} - Q(k \to 0) \sim \mathcal{O}(\epsilon_1)$ as explained above.

Finally, another problem arises when dealing with inhomogeneities large enough to form a PBH, namely, non-linear or even non-perturbative effects can play an important role, reason why we need to go beyond linear perturbation theory. The rest of the review is devoted to the study of inhomogeneities generated during inflation beyond linear perturbation theory.

## 5. Gradient Expansion

The gradient expansion is a non-perturbative, in terms of the amplitude of the inhomogeneities, expansion of the ADM equations valid when the characteristic wavelength of the inhomogeneities $L$ is much larger that the Hubble horizon $H_l^{-1}$ [100–109]. Since this is the same approximation that we do when studying the long wavelength behaviour of the linear perturbations, we will follow the same idea and we will take advantage of the fact that, when $L \gg H_l^{-1}$, the universe is locally homogeneous and isotropic to define an expansion parameter $\sigma \equiv \frac{H_l^{-1}}{L}$ such that at leading order in $\sigma$, each local patch of the universe of size $(\sigma H_l^{-1})$ (we will call this scale the coarse-grained scale) is approximately described by a FLRW universe. Higher order terms in the $\sigma$ will instead capture local inhomogeneities.

Contrary to the linear theory approach to cosmological perturbations, the gradient expansion is valid for any amplitude of local over-densities as long as the patch is taken small enough for the gradients to be negligible. Note that this assumption on which the gradient expansion is based on implies that a patch can be found such that any spatial gradient would only introduce an order $\sigma$. In other words, for any generic function $X$, $\partial_i X \sim X \times \mathcal{O}(\sigma)$. This is because a function which is approximately homogeneous in local coordinates can be written as $X(t, \sigma x^i)$ with $\sigma \ll 1$. Thus, we have

$$\partial_i X(t, \sigma x^i) = \sigma \frac{\partial}{\partial(\sigma x^i)} X(t, \sigma x^i) = \sigma \frac{\partial}{\partial(\sigma x^i)} X(t, \sigma x^i)\Big|_{\sigma x^i = 0} + \mathcal{O}(\sigma^2),$$

and since $\frac{\partial}{\partial(\sigma x^i)} X(t, \sigma x^i)\big|_{\sigma x^i = 0}$ can be of the same order as $X(t, \sigma x^i)$ we can generically write:

$$\partial_i X \sim X \times \mathcal{O}(\sigma).$$

When we were studying linear perturbation theory, the way of relating different patches was to define a global background over which each patch represented a perturbation. In this case we do not have a physical background metric because we are no longer perturbing anything; however we will still define a fictitious global background metric with coordinates $t$ and $x^i$, i.e.,

$$ds_b^2 = -dt^2 + a(t)^2 \delta_{ij} dx^i dx^j \,. \tag{82}$$

We are then interested in writing each local FLRW patch, which at leading order in $\sigma$ and in terms of local coordinates is simply (63), in terms of the coordinates $t$ and $x^i$. At leading order in gradient expansion and considering only scalar perturbations (note that, if we want to study inflation in a fully non-perturbative way, we should also take into account vector and tensor perturbations. This is because, although they are independent at linear order in perturbation theory, this is no longer true at higher orders. The reason why we do not include vector and tensor perturbations here is because, although at this level it would be straightforward, it is not possible when applying gradient expansion to stochastic inflation as we will see) we have:

$$ds_l^2 = - {}_{(0)}\alpha^2 dt^2 + {}_{(0)}\gamma_{ij}\left(dx^i + {}_{(0)}\beta^i dt\right)\left(dx^j + {}_{(0)}\beta^j\right), \tag{83}$$

with the conditions:

1. ${}_{(0)}\alpha = {}_{(0)}\alpha(t)$,
2. ${}_{(0)}\beta^i = b(t)x^i$,
3. ${}_{(0)}\gamma_{ij} = \gamma(t)\delta_{ij} = a(t)^2 e^{2\,{}_{(0)}\zeta(t)}\delta_{ij}$,

where we are using the subscript ${}_{(0)}$ to remind the reader that we are at leading order in gradient expansion It is important to remark here that the leading order in gradient expansion of each quantity can be different, for example, the leading order for $\alpha$ and $\zeta$ is $\mathcal{O}(\sigma^0)$ whereas the leading order for $\beta^i$ is $\mathcal{O}(\sigma^{-1})$. Having a quantity whose leading order is gradient expansion is $\mathcal{O}(\sigma^{-1})$ could seem problematic; however this is only telling us that $\beta^i$ is generically a non-local quantity, in fact, its linearization give us the non-local variable $B$ (see (23)) studied in Section 4.1.1. Furthermore, we will see that $\beta^i$ always appear together with a spatial derivative in the equations of motion.

One could be worried about the fact that a homogeneous and isotropic metric contains terms outside the diagonal; however, following [110] we know that a space-time is homogeneous and isotropic if:

1. All constant time hypersurfaces $\Sigma_t$ are constant curvature spaces. In our case the hypersurfaces $\Sigma_t$ are simply Euclidean and this condition is trivially satisfied.
2. The extrinsic curvature of the hypersurfaces is homogeneous and isotropic. Using the definition of extrinsic curvature of (12) together with the conditions for ${}_{(0)}\alpha$, ${}_{(0)}\beta^i$ and ${}_{(0)}\gamma_{ij}$ specified below (83), we can see that the extrinsic curvature only depends on time and hence this condition is also satisfied.

Different functions for ${}_{(0)}\alpha$, ${}_{(0)}\beta^i$ and ${}_{(0)}\zeta$ will give different FLRW patches as long as they satisfy the conditions given below (83). We can then relate the different locally homogeneous and isotropic patches by knowing the different non-perturbative functions for ${}_{(0)}\alpha$, ${}_{(0)}\beta^i$ and ${}_{(0)}\zeta$ that lead to each one of them. Of course, in the same way as in perturbation theory, the value of ${}_{(0)}\alpha$, ${}_{(0)}\beta^i$ and ${}_{(0)}\zeta$ will depend on the gauge choice and on the solution for the ADM equations.

We can generalize the idea of writing a FLRW metric in terms of $x^i$ and $t$ to writing quasi-FLRW metrics using the same coordinates. Following the ADM formalism we can write a metric valid at all orders in gradient expansion. With the identification $\gamma_{ij} = a(t)^2 e^{2\zeta}\tilde{\gamma}_{ij}$ we have:

$$ds^2 = g_{\mu\nu}dx^\mu dx^\nu = -\alpha^2 dt^2 + a(t)^2 e^{2\zeta}\tilde{\gamma}_{ij}(dx^i + \beta^i dt)(dx^j + \beta^j dt)\,, \tag{84}$$

where the leading order in gradient expansion for each variable is:

$$_{(0)}\alpha \sim \mathcal{O}(\sigma^0)\,, \qquad _{(0)}\zeta \sim \mathcal{O}(\sigma^0)\,, \qquad _{(0)}\beta^i \sim \mathcal{O}(\sigma^{-1})\,,$$
$$\tilde{\gamma}_{ij} - \delta_{ij} \sim \mathcal{O}(\sigma)\,, \qquad _{(0)}\phi \sim \mathcal{O}(\sigma^0)\,. \tag{85}$$

The last term has been added to take into account the expansion of the scalar field, which is generically non-zero at the background level. It is important to realize here that the condition $\tilde{\gamma}_{ij} - \delta_{ij} \sim \mathcal{O}(\sigma)$ implies a further condition on the scalar part of $\tilde{\gamma}_{ij}$, in fact, using the expansion of the exponential of a matrix we can write

$$\tilde{\gamma}_{ij} = e^{-2M_{ij}} \simeq \delta_{ij} - 2M_{ij} + \mathcal{O}(\sigma^2)\,. \tag{86}$$

Now, $M_{ij}$ must be traceless by definition (see see the paragraph under (23)). Focusing only in the scalar part of $M_{ij}$ we can then write

$$\tilde{\gamma}_{ij} - \delta_{ij} \simeq -2\left(\partial_i\partial_j - \frac{1}{3}\delta_{ij}\nabla^2\right)C + \mathcal{O}(\sigma^2)\,, \tag{87}$$

where $C$ is a scalar function. This immediataly implies that $\left(\partial_i\partial_j - \frac{1}{3}\delta_{ij}\nabla^2\right)C \sim \mathcal{O}(\sigma)$. Note that, as we will see later on, this condition is not in contradiction with $\nabla^2 C \sim \mathcal{O}(\sigma^0)$.

*5.1. $\mathcal{O}(\sigma^0)$ in Gradient Expansion*

Having specified the leading order gradient expansion for each one of the non-perturbative variables, we can now expand the ADM equations in terms of $\sigma$. As an example we will expand the Hamiltonian constraint at $\mathcal{O}(\sigma^0)$ using the spatially flat gauge, i.e.,

$$(\gamma_{\mathrm{f}})_{ij} = a^2 \delta_{ij}\,, \tag{88}$$

where the subscript f stands for "flat". Since in this gauge we also have $R_{\mathrm{f}}^{(3)} = 0$ we can write the Hamiltonian constraint from (14) as:

$$-(A_{\mathrm{f}})_{ij}(A_{\mathrm{f}})^{ij} + \frac{2}{3}K_{\mathrm{f}}^2 - \frac{2}{M_{PL}^2}(T_{\mathrm{f}})_{\mu\nu}(n_{\mathrm{f}})^\mu (n_{\mathrm{f}})^\nu = 0\,, \tag{89}$$

where $(n_{\mathrm{f}})^\mu = g^{\mu\nu}(n_{\mathrm{f}})_\nu = \left(\frac{1}{\alpha_{\mathrm{f}}}, -\frac{(\beta_{\mathrm{f}})^i}{\alpha_{\mathrm{f}}}\right)$.

Equation (89) can be written in terms of the metric variables $\alpha_{\mathrm{f}}$ and $(\beta_{\mathrm{f}})^i$ of (84) and the scalar field $\phi_{\mathrm{f}}$. Using the results for $(A_{\mathrm{f}})_{ij}$ and $K_{\mathrm{f}}$ given by (18) and (17) respectively we have:

$$-\frac{1}{4\alpha_{\mathrm{f}}^2}\left[\delta^{ik}\partial^j(\beta_{\mathrm{f}})_k + \delta^{jk}\partial^i(\beta_{\mathrm{f}})_k - \frac{2}{3}\delta^{ij}\partial^k(\beta_{\mathrm{f}})_k\right]\left[\delta_{ik}\partial_j(\beta_{\mathrm{f}})^k + \delta_{jk}\partial_i(\beta_{\mathrm{f}})^k - \frac{2}{3}\delta_{ij}\partial_k(\beta_{\mathrm{f}})^k\right]$$
$$+\frac{2}{3}\left(-3\frac{H^b}{\alpha_{\mathrm{f}}} + \frac{1}{\alpha_{\mathrm{f}}}\partial_k(\beta_{\mathrm{f}})^k\right)^2$$
$$-\frac{2}{M_{PL}^2}\left[\frac{\dot{\phi}_{\mathrm{f}}^2}{2\alpha_{\mathrm{f}}^2} - \frac{\dot{\phi}_{\mathrm{f}}(\beta_{\mathrm{f}})^i\partial_i\phi_{\mathrm{f}}}{\alpha_{\mathrm{f}}^2} + \frac{(\beta_{\mathrm{f}})^i(\beta_{\mathrm{f}})^j\partial_i\phi_{\mathrm{f}}\partial_j\phi_{\mathrm{f}}}{2\alpha_{\mathrm{f}}^2} + \frac{\partial^i\phi_{\mathrm{f}}\partial_i\phi_{\mathrm{f}}}{2a^2} + V(\phi_{\mathrm{f}})\right] = 0\,, \tag{90}$$

which is valid at all orders in gradient expansion. Keeping only the $\mathcal{O}(\sigma^0)$ terms we get:

$$\frac{2}{3}\left(-3\frac{H^b}{{}_{(0)}\alpha_{\mathrm{f}}} + \frac{1}{\alpha_{\mathrm{f}}}\partial_k\left({}_{(0)}\beta_{\mathrm{f}}\right)^k\right)^2$$

$$-\frac{2}{M_{PL}^2}\left[\frac{\left({}_{(0)}\dot{\phi}_{\mathrm{f}}\right)^2}{2\left({}_{(0)}\alpha_{\mathrm{f}}\right)^2} - \frac{{}_{(0)}\dot{\phi}_{\mathrm{f}}\left({}_{(0)}\beta_{\mathrm{f}}\right)^i{}_{(0)}(\partial_i\phi_{\mathrm{f}})}{\left({}_{(0)}\alpha_{\mathrm{f}}\right)^2} + \frac{\left({}_{(0)}\beta_{\mathrm{f}}\right)^i\left({}_{(0)}\beta_{\mathrm{f}}\right)^j{}_{(0)}(\partial_i\phi_{\mathrm{f}}){}_{(0)}(\partial_j\phi_{\mathrm{f}})}{2\left({}_{(0)}\alpha_{\mathrm{f}}\right)^2} + V(\phi_{\mathrm{f}})\right] = 0, \tag{91}$$

where by ${}_{(0)}(\partial_i\phi_{\mathrm{f}})$ we mean $\sigma\frac{\partial}{\partial(\sigma x^i)}\phi(t,\sigma x^i)\big|_{\sigma x^i=0}$ as explained before.

Note that the first line of (90) is $\mathcal{O}(\sigma)$ because of the condition of ${}_{(0)}\beta^i$ below Equation (83). Following this procedure, it is straightforward to write all the ADM equations at order $\mathcal{O}(\sigma^0)$; however, we know from linear perturbation theory that there are equations with global spatial derivatives that play a role in the $k \to 0$ limit, which translated to the gradient expansion way of thinking means that there are equations that contain an overall factor $\sigma$ and hence in order to extract its contribution at $\mathcal{O}(\sigma^0)$ they must be written up to $\mathcal{O}(\sigma)$. We know that the momentum constraint is one of these equations so let us analyze it. In spatially flat gauge, the momentum constraint (15) is:

$$\partial^j\left(\tilde{A}_{\mathrm{f}}\right)_{ij} - \frac{2}{3}\partial_i K_{\mathrm{f}} = \frac{1}{M_{PL}^2}J_i \tag{92}$$

If we expand (92) up to $\mathcal{O}(\sigma)$ we get:

$$-\frac{2}{3}{}_{(0)}\partial_i\left(-3\frac{H^b}{\alpha_{\mathrm{f}}}\right) = \frac{1}{M_{PL}^2}\left(-\frac{1}{{}_{(0)}\alpha_{\mathrm{f}}}{}_{(0)}\dot{\phi}_{\mathrm{f}}{}_{(0)}(\partial_i\phi_{\mathrm{f}}) + \frac{\left({}_{(0)}\beta_{\mathrm{f}}\right)^k}{{}_{(0)}\alpha_{\mathrm{f}}}{}_{(0)}(\partial_k\phi_{\mathrm{f}}){}_{(0)}(\partial_l\phi_{\mathrm{f}})\right). \tag{93}$$

An important aspect regarding the gradient expansion is worthy to remark at this point, note that, in the derivation of (93) we have used $\partial^j\left(\tilde{A}_{\mathrm{f}}\right)_{ij} \sim \mathcal{O}(\sigma)$, which seems in contradiction with $\left(\tilde{A}_{\mathrm{f}}\right)_{ij} \sim \mathcal{O}(\sigma)$ (as we saw when deriving the Hamiltonian constraint at $\mathcal{O}(\sigma^0)$) and the fact that each spatial gradient introduces an order in $\sigma$. However, in this case we have an exception due to the traceless nature of $\left(\tilde{A}_{\mathrm{f}}\right)_{ij}$. Let us see why: from the condition 2 below (83) we have $(\beta_{\mathrm{f}})^i \simeq b(t,\sigma x^l)x^i$ and from (18) in spatially flat gauge:

$$\partial^j\left(\tilde{A}_{\mathrm{f}}\right)_{ij} = \delta_{ik}\partial_j\partial^j(\beta_{\mathrm{f}})^k + \partial_i\partial_k(\beta_{\mathrm{f}})^k - \frac{2}{3}\partial_i\partial_k(\beta_{\mathrm{f}})^k = \frac{4}{3}\partial_i\partial_k(\beta_{\mathrm{f}})^k = 4\partial_i b(t,\sigma x^l) + \mathcal{O}(\sigma^2),$$

which is clearly $\mathcal{O}(\sigma)$.

The linear version of (93) obviously corresponds to the linear momentum constraint in spatially flat gauge, i.e.,

$$\partial_i\left(H^b A_{\mathrm{f}}\right) = \partial_i\left(\frac{\dot{\phi}^b}{2M_{PL}^2}\delta\phi_{\mathrm{f}}\right). \tag{94}$$

As already stressed during Section 4.1.1, although (94) is an equation that appears at $\mathcal{O}(\sigma)$, it contains information at $\mathcal{O}(\sigma^0)$, namely

$$H^b A_{\mathrm{f}} = \frac{\dot{\phi}^b}{2M_{PL}^2}\delta\phi_{\mathrm{f}}. \tag{95}$$

Unfortunately, things are not that easy when dealing with the fully non-perturbative momentum constraint at $\mathcal{O}(\sigma)$, in fact, if we look at (93), we will easily realize that the total spatial derivative is no longer present. How the $\mathcal{O}(\sigma^0)$ information is encoded in (93) in a fully non-perturbative way is beyond the scope of this review, neverthe-

less, it is very important to be aware that we must take this information into account if we want to correctly describe the non-perturbative and long wavelength dynamics of the inhomogeneities, otherwise we would be making a mistake equivalent as when using the linear separate universe approach of Section 4.1.2 instead of the $k \to 0$ limit of Section 4.1.1. Nevertheless, due to its usefulness, in the following section we will present the non-perturbative version of the linear separate universe approach.

*5.2. Non-Perturbative Separate Universe Approach*

The $\mathcal{O}(\sigma^0)$ expression for the Hamiltonian constraint (91) seems very difficult to solve, and we have just argued the difficulties that arise when trying to extract the $\mathcal{O}(\sigma^0)$ information from the $\mathcal{O}(\sigma)$ momentum constraint. This is why a further approximation in addition to the $\mathcal{O}(\sigma^0)$ gradient expansion is usually performed in the literature [20,111,112]. This new approximation is the separate universe approach, which assumes that each FLRW patch of the universe evolve independently one from each other. We have already studied the linear version of the separate universe approach in Section 4.1.2, where we have seen that this assumption implies that neither non-local terms nor the momentum constraint will be present, which is equivalent to state that $\beta^i \sim \mathcal{O}(\sigma^3)$ as done in [109]. Within this approximation, the Hamiltonian constraint in spatially flat gauge of (91) takes a much simpler form:

$$\left( \frac{H^b}{{}_{(0)}\alpha_f^{sep}} \right)^2 = \frac{1}{3M_{PL}^2} \left[ \frac{\left( {}_{(0)}\dot{\phi}_f^{sep} \right)^2}{2 \left( {}_{(0)}\alpha_f^{sep} \right)^2} + V\left( {}_{(0)}\phi_f^{sep} \right) \right], \tag{96}$$

which reminds us of the background Hamiltonian constraint (4). In fact, this coincidence extends also to the equation of motion of the scalar field as one could expect since we are assuming every FLRW to evolve independently from the others. A famous application of the separate universe approach is the $\delta N$ formalism in order to compute the uniform-density curvature perturbation in a non perturbative way [70–73]. As a reminder, the linear uniform-density curvature perturbation is defined as

$$\mathcal{R}_{ud} \equiv -\frac{H^b}{\rho^b} \delta\rho_f,$$

where $\rho$ is the energy density of the scalar field, which in inflation is $\rho = 3M_{PL}^2 \left( H^b \right)^2$ and $\delta\rho_f$ is the perturbation of the energy density in spatially flat gauge.

In this section we have presented the non-perturbative generalization of the long wavelength linear perturbation theory presented in Section 4, being the $\mathcal{O}(\sigma^0)$ in gradient expansion of Section 5.1 the generalization of the $k \to 0$ limit for scalar perturbations of Section 4.1.1 and being the separate universe approach the non-linear generalization of the $k = 0$ (or linear separate universe approach) case of Section 4.1.2. For this reason, it is obvious that the $\mathcal{O}(\sigma^0)$ gradient expansion and the separate universe approach will give different "decaying" terms, being the ones of the $\mathcal{O}(\sigma^0)$ gradient expansion the correct ones, as it happened in linear theory.

Using the results from linear theory again, we can conclude that the error made when using the separate universe approach instead of the $\mathcal{O}(\sigma^0)$ gradient expansion will be generically of $\mathcal{O}(\epsilon_1)$, strongly depending on the inflationary regime. This conclusion is deduced of course assuming that higher orders in perturbation theory will not spoil the results at leading order, assumption that can be problematic if the inhomogeneity under study is non-perturbative.

We will finish this section by reminding the reader that setting initial conditions for long wavelength perturbations is problematic because we cannot use the Bunch-Davies vacuum, which is only well defined when $\frac{k}{aH} \to \infty$. This is why, although the gradient expansion is a very useful way to study inhomogeneities in a non-perturbative way, we need some other tool to set the initial conditions.

## 6. Stochastic Approach to Inflation

The stochastic approach to inflation combines the two approximations schemes presented until now to study the evolution of inhomogeneities in a non-perturbative way. The idea is to split the variables of interest (let us say *X*) into two parts: an infrared (IR) part that contains all the inhomogeneities with characteristic wavelength larger that some coarse-grained scale $(\sigma H)^{-1}$ ($\sigma$ is the same parameter as the one used in gradient expansion) and a ultraviolet (UV) part, which encompasses inhomogeneities with characteristic scale smaller than $(\sigma H)^{-1}$ (or characteristic wavenumber *k* bigger than $\sigma aH$).

Since the UV part starts evolving well inside the Hubble horizon, we will assume that it is perturbatively small. Thus, one can use linear perturbation theory to describe it, where initial conditions are well defined. The IR part instead can be large; however, since the IR part only contains long wavelengths, the gradient expansion can be used there. As we will see, whenever an UV mode exits the coarse-grained scale, it will act as a kick for the IR part, solving the initial condition problem of gradient expansion. Note that, although we will only study stochastic inflation in the context of Einstein's gravity, the only requirements for the construction of a stochastic formalism are the possibility of a separation between IR and UV modes and a well-behaved perturbative expansion. There is then no reason a priori to think that the stochastic framework can not be applied to modifications of Einstein's gravity such as massive gravity [113,114].

The stochastic formalism is then a mathematical framework that, in principle, allows us to study the inhomogeneities generated during inflation in a non-perturbative way, reason why it is widely used when studying PBHs formation (see for example [115,116]). To see how stochastic inflation works and why it is called "stochastic" we will derive the formalism step by step. From Section 5 it should be clear now that we will have different stochastic equations depending on if we are using the separate universe approach or the $\mathcal{O}(\sigma^0)$ gradient expansion. The stochastic formalism that uses the separate universe approach is the most widely used in the literature (see for example [10,14–19,23,27,35,37,38, 51–53,63,66,115,116]) so we will start with its derivation. After that, we will take advantage of the stochastic equations just derived to present a stochastic formalism based on the $\mathcal{O}(\sigma^0)$ gradient expansion [65].

Before starting with the derivation and, since it is not as trivial as the spatially flat gauge, let us present the gauge we will be working with: the uniform-N gauge. We define the number of e-folds *N* as the integrated expansion rate of $\Sigma_t$ hypersurfaces, i.e.,

$$N \equiv -\frac{1}{3} \int K dt_l \,, \tag{97}$$

where $K \equiv -3H_l$ (being $H_l$ the local Hubble parameter) is the extrinsic curvature defined in (12), which coincides with the expansion rate of $\Sigma_t$ hypersurfaces as shown in Section 3 and $t_l$ is the local time of (63). In terms of the variables and coordinates of the ADM metric (84) *N* can be written as:

$$N \equiv \int \left( H^b + \dot{\zeta} - \frac{1}{3} D_i \beta^i \right) dt \,. \tag{98}$$

The uniform-N gauge is defined such that *N* coincides with the number of e-folds defined in the global background of (82), i.e., $N \equiv \int H^b dt$. From (98), this immediately implies $\zeta_{\delta N} = 0$ and $(\beta_{\delta N})^i = 0$ (where the subscript $\delta N$ specifies the gauge), or $D_{\delta N} = 0$ and $B_{\delta N} = 0$ in its linear limit (see Equation (23)). The reason behind this gauge choice will be clarified along the following two subsections.

### 6.1. Stochastic Formalism Based on the Separate Universe Approach

As we have already indicated, this is the stochastic formalism most widely used in the literature due to its simplicity; however, as we will see, it has some problems. First of all, it is very important to realize that in the separate universe approach, both the uniform-N gauge and the spatially flat gauge are equivalent, which leads to many authors to use the

uniform-N gauge for the IR part and the spatially flat gauge for the UV part [60]. Let us see why:

As we know, in the separate universe approach, due to the absence of non-local terms, both $(\tilde{\gamma}^{sep})_{ij} = \delta_{ij}$ and $\partial_i(\beta^{sep})^i = 0$ are automatically satisfied (see (66)). This means that, under this assumption, we have to add the condition $\partial_i\left(\beta_{\rm f}^{sep}\right)^i = 0$ to the spatially flat gauge where $(\tilde{\gamma}_{\rm f}) = \delta_{ij}$ and $\zeta_{\rm f} = 0$. In the same way, we have to add the condition $\left(\tilde{\gamma}_{\delta N}^{sep}\right)_{ij} = \delta_{ij}$ to the uniform-N gauge, where $\partial_i(\beta_{\delta N})^i = 0$ and $\zeta_{\delta N} = 0$. The main consequence of this is that, under the separate universe condition, we can express all the scalar fluctuations only in terms of the field inhomogeneities not only in the spatially flat gauge, but also in the uniform-N gauge. This can be clearly seen when looking at the linear "separate-universe" gauge invariant MS variable of (68). For non-linear generalizations of this variables see [117,118].

Before continuing, let us remind the reader that the equivalence between these two gauges in only valid under the separate universe approach, which, as seen in Section 4.1.2, it generically fails at $\mathcal{O}(\epsilon_1)$.

During this section, and although it is not compulsory, we will work using the number of e-folds $N \equiv \int H^b dt$ as time variable (Note that we can use any time variable we want because we will use the coordinates of a fictitious global background, i.e., the coordinates of (84). If we would instead use local coordinates (as in (63)), we would be interested in using an unperturbed time variable, being $N$ the natural choice in the uniform-N gauge). Another important aspect is that, in order not to overload notation, we will suppress the subscript $\delta N$ indicating the gauge we are using, such that in the following, unless otherwise stated, a variable without a subscript that indicates the gauge is a variable in the uniform-N gauge.

As indicated before, the stochastic formalism presented in this subsection is based on the separate universe approach for the IR part and on linear perturbation theory for the UV part. To illustrate this we will consider in detail the equation of motion for the trace the extrinsic curvature (19) in uniform-N gauge, which can be written using the variables of the ADM metric (84) as:

$$-3\frac{H^b}{\alpha}\frac{\partial}{\partial N}\left(\frac{H^b}{\alpha}\right) - \left(\frac{H^b}{2\alpha}\right)^2\frac{\partial\tilde{\gamma}_{ij}}{\partial N}\frac{\partial\tilde{\gamma}^{ij}}{\partial N} - 3\left(\frac{H^b}{\alpha}\right)^2 + D_k D^k\alpha - \frac{1}{M_{PL}^2}\left(\left(\frac{H^b}{\alpha}\right)^2\left(\frac{\partial\phi}{\partial N}\right)^2 - V(\phi)\right) = 0. \qquad (99)$$

The first thing to do is to split the variables of interest into their IR and UV part. In this case we only have two variables to split:

$$\begin{aligned} \alpha &= \alpha^{IR} + \alpha^{UV}, \\ \phi &= \phi^{IR} + \phi^{UV}. \end{aligned} \qquad (100)$$

In (100) we are not considering $\frac{\partial\tilde{\gamma}_{ij}}{\partial N}$ as a variable of interest not only because $\tilde{\gamma}_{ij} = \delta_{ij}$ in the separate universe approach, but also because $\frac{\partial\tilde{\gamma}_{ij}}{\partial N}\frac{\partial\tilde{\gamma}^{ij}}{\partial N} \sim \mathcal{O}(\sigma^2)$ in gradient expansion and quadratic in perturbation theory so it does not play any role even if we were using $\mathcal{O}(\sigma^0)$ gradient expansion.

Due to the perturbative nature of the UV variables, we will expand (99) keeping only linear terms in UV and isolate them in the right hand side of the equation getting

$$- 3\frac{H^b}{\alpha^{IR}}\frac{\partial}{\partial N}\left(\frac{H^b}{\alpha^{IR}}\right) - 3\left(\frac{H^b}{\alpha^{IR}}\right)^2 + D^k D_k \alpha^{IR} - \frac{1}{M_{PL}^2}\left(\left(\frac{H^b}{\alpha^{IR}}\right)^2\left(\frac{\partial\phi^{IR}}{\partial N}\right)^2 - V\left(\phi^{IR}\right)\right)$$

$$= -3\frac{\left(H^b\right)^2}{(\alpha^{IR})^3}\frac{\partial\alpha^{UV}}{\partial N} + \left(\frac{9\left(H^b\right)^2}{(\alpha^{IR})^4}\frac{\partial\alpha^{IR}}{\partial N} - \frac{6H^b}{(\alpha^{IR})^3}\left(\frac{\partial H^b}{\partial N}\right)\right)\alpha^{UV} - \frac{6\left(H^b\right)^2}{(\alpha^{IR})^3}\alpha^{UV} - \frac{\nabla^2}{a^2}\alpha^{UV}$$

$$+ \frac{1}{M_{PL}^2}\left[2\left(\frac{H^b}{\alpha^{IR}}\right)^2\frac{\partial\phi^{IR}}{\partial N}\frac{\partial\phi^{UV}}{\partial N} - 2\frac{\left(H^b\right)^2}{(\alpha^{IR})^3}\left(\frac{\partial\phi^{IR}}{\partial N}\right)^2\alpha^{UV} - V_\phi\left(\phi^{IR}\right)\phi^{UV}\right]. \tag{101}$$

Note that this is the same as we did in linear perturbation theory of Section 4 but using the metric (63) (or equivalently (84)) as background metric.

Now, since the IR variables are well outside the Hubble horizon, we will use the separate universe approach for them. Since $\alpha^{IR} \sim \mathcal{O}(\sigma^0)$ and hence $D_k D^k \alpha^{IR} \sim \mathcal{O}(\sigma^2)$ we have:

$$- 3\frac{H^b}{{}_{(0)}\alpha^{IR}}\frac{\partial}{\partial N}\left(\frac{H^b}{{}_{(0)}\alpha^{IR}}\right) - 3\left(\frac{H^b}{{}_{(0)}\alpha^{IR}}\right)^2 - \frac{1}{M_{PL}^2}\left(\left(\frac{H^b}{{}_{(0)}\alpha^{IR}}\right)^2\left(\frac{\partial {}_{(0)}\phi^{IR}}{\partial N}\right)^2 - V\left({}_{(0)}\phi^{IR}\right)\right)$$

$$= -3\frac{\left(H^b\right)^2}{\left({}_{(0)}\alpha^{IR}\right)^3}\frac{\partial\alpha^{UV}}{\partial N} + \left(\frac{9\left(H^b\right)^2}{\left({}_{(0)}\alpha^{IR}\right)^4}\frac{\partial {}_{(0)}\alpha^{IR}}{\partial N} - \frac{6H^b}{\left({}_{(0)}\alpha^{IR}\right)^3}\left(\frac{\partial H^b}{\partial N}\right)\right)\alpha^{UV} - \frac{6\left(H^b\right)^2}{\left({}_{(0)}\alpha^{IR}\right)^3}\alpha^{UV}$$

$$- \frac{\nabla^2}{a^2}\alpha^{UV} + \frac{1}{M_{PL}^2}\left[2\left(\frac{H^b}{{}_{(0)}\alpha^{IR}}\right)^2\frac{\partial {}_{(0)}\phi^{IR}}{\partial N}\frac{\partial\phi^{UV}}{\partial N} - 2\frac{\left(H^b\right)^2}{\left({}_{(0)}\alpha^{IR}\right)^3}\left(\frac{\partial {}_{(0)}\phi^{IR}}{\partial N}\right)^2\alpha^{UV} - V_\phi\left({}_{(0)}\phi^{IR}\right)\phi^{UV}\right], \tag{102}$$

where we have inserted an extra subindex ${}_{(0)}$ to indicate that we are at leading order in gradient expansion.

Using Fourier analysis we can now define more rigorously the IR and UV modes. If we choose the Heaviside theta as a window function (The choice of the Heaviside theta as window function in the stochastic formalism is the most common one. However it can lead to some problems as indicated in [28]) we have the following decomposition for a generic function X.

$$X^{IR}(t,\mathbf{x}) \equiv \int \frac{d\mathbf{k}}{(2\pi)^{3/2}}\Theta(\sigma a_l(N)H_l(N) - k)\mathcal{X}_\mathbf{k}^{IR}(t,\mathbf{x}),$$

$$X^{UV}(t,\mathbf{x}) \equiv \int \frac{d\mathbf{k}}{(2\pi)^{3/2}}\Theta(k - \sigma a_l(N)H_l(N))\mathcal{X}_\mathbf{k}^{UV}(t,\mathbf{x}), \tag{103}$$

where, similarly as in linear perturbation theory (see (47)),$\mathcal{X}_\mathbf{k}^{UV}(t,\mathbf{x})$ is define as the following hermitian operator:

$$\mathcal{X}_\mathbf{k}^{UV}(t,\mathbf{x}) = e^{-i\mathbf{k}\cdot\mathbf{x}}X_\mathbf{k}(N)a_\mathbf{k} + e^{i\mathbf{k}\cdot\mathbf{x}}X_\mathbf{k}^* a_\mathbf{k}^\dagger, \tag{104}$$

where $X_\mathbf{k}(N)$ is the solution of the evolution equation for the perturbation X over the local background defined by (83) and $a_\mathbf{k}$ and $a_\mathbf{k}^\dagger$ are the usual creation and annihilation operators which follow the commutation relation given in (48).

Note that, in the spirit of gradient expansion, the splitting is done in the local cosmological coarse-grained scale $(\sigma H_l)^{-1}$, which generically differs form the one of the background,

for example in uniform-N gauge we have $H_l = \frac{H^b}{{}_{(0)}\alpha^{IR}}$. Inserting the definition of $X^{UV}$ of (103) into (102) we get:

$$-3\frac{H^b}{{}_{(0)}\alpha^{IR}}\frac{\partial}{\partial N}\left(\frac{H^b}{{}_{(0)}\alpha^{IR}}\right) - 3\left(\frac{H^b}{{}_{(0)}\alpha^{IR}}\right)^2 - \frac{1}{M_{PL}^2}\left(\left(\frac{H^b}{{}_{(0)}\alpha^{IR}}\right)^2\left(\frac{\partial\,_{(0)}\phi^{IR}}{\partial N}\right)^2 - V\left({}_{(0)}\phi^{IR}\right)\right)$$

$$3\frac{\left(H^b\right)^2}{\left({}_{(0)}\alpha^{IR}\right)^3}\frac{\partial}{\partial N}\left(\sigma a\frac{H^b}{{}_{(0)}\alpha^{IR}}\right)\int\frac{d\mathbf{k}}{(2\pi)^{3/2}}\delta\left(k - \sigma a\frac{H^b}{{}_{(0)}\alpha^{IR}}\right)\alpha_{\mathbf{k}}^{UV}$$

$$-\frac{2}{M_{PL}^2}\left(\frac{H^b}{{}_{(0)}\alpha^{IR}}\right)^2\frac{\partial\,_{(0)}\phi^{IR}}{\partial N}\frac{\partial}{\partial N}\left(\sigma a\frac{H^b}{{}_{(0)}\alpha^{IR}}\right)\int\frac{d\mathbf{k}}{(2\pi)^{3/2}}\delta\left(k - \sigma a\frac{H^b}{{}_{(0)}\alpha^{IR}}\right)\varphi_{\mathbf{k}}^{UV}$$

$$+\int\frac{d\mathbf{k}}{(2\pi)^{3/2}}\Theta\left(k - \sigma a\frac{H^b}{{}_{(0)}\alpha^{IR}}\right)\left\{-3\frac{\left(H^b\right)^2}{\left({}_{(0)}\alpha^{IR}\right)^3}\frac{\partial\alpha_{\mathbf{k}}^{UV}}{\partial N}\right.$$

$$+\left(\frac{9\left(H^b\right)^2}{\left({}_{(0)}\alpha^{IR}\right)^4}\frac{\partial\,_{(0)}\alpha^{IR}}{\partial N} - \frac{6H^b}{\left({}_{(0)}\alpha^{IR}\right)^3}\left(\frac{\partial H^b}{\partial N}\right)\right)\alpha_{\mathbf{k}}^{UV} - \frac{6\left(H^b\right)^2}{\left({}_{(0)}\alpha^{IR}\right)^3}\alpha_{\mathbf{k}}^{UV} + \frac{k^2}{a^2}\alpha_{\mathbf{k}}^{UV}$$

$$\frac{1}{M_{PL}^2}\left[2\left(\frac{H^b}{{}_{(0)}\alpha^{IR}}\right)^2\frac{\partial\,_{(0)}\phi^{IR}}{\partial N}\frac{\partial\varphi_{\mathbf{k}}^{UV}}{\partial N} - \frac{2\left(H^b\right)^2}{\left({}_{(0)}\alpha^{IR}\right)^3}\left(\frac{\partial\,_{(0)}\phi^{IR}}{\partial N}\right)^2\alpha_{\mathbf{k}}^{UV} - V_\phi\left({}_{(0)}\phi^{IR}\right)\varphi_{\mathbf{k}}^{UV}\right]\right\}, \tag{105}$$

where $\alpha_{\mathbf{k}}^{UV}$ and $\varphi_{\mathbf{k}}^{UV}$ are operators defined as in (104).

The right-hand side of (105) has two different terms:

1. The second integral (terms multiplying the Heaviside theta) is the evolution equation for the extrinsic curvature linearized over a local FLRW patch defined by ${}_{(0)}\alpha^{IR}$ and ${}_{(0)}\phi^{IR}$. Once the Bunch-Davies vacuum is chosen for that patch, this term will be automatically satisfied so it can be consistently set to zero. Note that the solution of this part equalized to zero is precisely what give us the functions $X_{\mathbf{k}}$ in (104).

2. The first two integrals, proportional to a Dirac delta, can be seen as boundary conditions and hence they will act as the initial conditions missing when using only gradient expansion.

We then get:

$$-3\frac{H^b}{{}_{(0)}\alpha^{IR}}\frac{\partial}{\partial N}\left(\frac{H^b}{{}_{(0)}\alpha^{IR}}\right) - 3\left(\frac{H^b}{{}_{(0)}\alpha^{IR}}\right)^2 - \frac{1}{M_{PL}^2}\left(\left(\frac{H^b}{{}_{(0)}\alpha^{IR}}\right)^2\left(\frac{\partial\,_{(0)}\phi^{IR}}{\partial N}\right)^2 - V\left({}_{(0)}\phi^{IR}\right)\right)$$

$$= -3\frac{\left(H^b\right)^2}{\left({}_{(0)}\alpha^{IR}\right)^3}\xi_3 + \frac{2}{M_{PL}^2}\left(\frac{H^b}{{}_{(0)}\alpha^{IR}}\right)^2\frac{\partial\,_{(0)}\phi^{IR}}{\partial N}\xi_1, \tag{106}$$

where we have defined $\xi_1$ and $\xi_3$ as:

$$\xi_1 \equiv -\frac{\partial}{\partial N}\left(\sigma a\frac{H^b}{{}_{(0)}\alpha^{IR}}\right)\int\frac{d\mathbf{k}}{(2\pi)^{3/2}}\delta\left(k - \sigma a\frac{H^b}{{}_{(0)}\alpha^{IR}}\right)\varphi_{\mathbf{k}}^{UV},$$

$$\xi_3 \equiv -\frac{\partial}{\partial N}\left(\sigma a\frac{H^b}{{}_{(0)}\alpha^{IR}}\right)\int\frac{d\mathbf{k}}{(2\pi)^{3/2}}\delta\left(k - \sigma a\frac{H^b}{{}_{(0)}\alpha^{IR}}\right)\alpha_{\mathbf{k}}^{UV}. \tag{107}$$

For the moment we will not characterize the quantities $\xi_1$ and $\xi_3$. If we now follow the procedure for the evolution equation of the trace of the extrinsic curvature just explained with the Hamiltonian constraint we get:

$$6\left(\frac{H^b}{_{(0)}\alpha^{IR}}\right)^2 - \frac{2}{M_{PL}^2}\left[\left(\frac{H^b}{_{(0)}\alpha^{IR}}\right)^2\left(\frac{\partial_{(0)}\phi^{IR}}{\partial N}\right)^2 + V\left(_{(0)}\phi^{IR}\right)\right] = \frac{2}{M_{PL}^2}\left(\frac{H^b}{_{(0)}\alpha^{IR}}\right)^2\frac{\partial_{(0)}\phi^{IR}}{\partial N}\xi_1, \qquad (108)$$

which can be solved for $\left(\frac{H^b}{_{(0)}\alpha^{IR}}\right)^2$, i.e.,

$$\left(\frac{H^b}{_{(0)}\alpha^{IR}}\right)^2 = \frac{V\left(_{(0)}\phi^{IR}\right)}{3M_{PL}^2 - \frac{1}{2}\left(\frac{\partial_{(0)}\phi^{IR}}{\partial N}\right)^2 - \frac{\partial_{(0)}\phi^{IR}}{\partial N}\xi_1}. \qquad (109)$$

Finally, the stochastic equation of motion for the field is obtained in the same way:

$$\frac{\partial^2_{(0)}\phi^{IR}}{\partial N^2} + \left(3 + \frac{\frac{\partial}{\partial N}\left(\frac{H^b}{_{(0)}\alpha^{IR}}\right)}{\frac{H^b}{_{(0)}\alpha^{IR}}}\right)\frac{\partial_{(0)}\phi^{IR}}{\partial N} + \frac{V_\phi\left(_{(0)}\phi^{IR}\right)}{\left(\frac{H^b}{_{(0)}\alpha^{IR}}\right)^2}$$

$$= -\frac{\partial\xi_1}{\partial N} - \xi_2 - \left(3 + \frac{\frac{\partial}{\partial N}\left(\frac{H^b}{_{(0)}\alpha^{IR}}\right)}{\frac{H^b}{_{(0)}\alpha^{IR}}}\right)\xi_1 + \frac{\partial_{(0)}\phi^{IR}}{\partial N}\frac{\xi_3}{_{(0)}\alpha^{IR}}, \qquad (110)$$

where $\xi_2$ is defined similarly to $\xi_1$ and $\xi_3$:

$$\xi_2 \equiv -\frac{\partial}{\partial N}\left(\sigma a\frac{H^b}{_{(0)}\alpha^{IR}}\right)\int\frac{d\mathbf{k}}{(2\pi)^{3/2}}\delta\left(k - \sigma a\frac{H^b}{_{(0)}\alpha^{IR}}\right)\frac{\partial\varphi_{\mathbf{k}}^{UV}}{\partial N}. \qquad (111)$$

As anticipated before, the usage and uniform-N gauge and the separate universe approach ensures that all the scalar inhomogeneities are encoded in the scalar field. This becomes clearer once we realize that we can write the system (106)–(110) in terms only of the scalar field. Inserting (106) and (109) into (110) and neglecting $\xi_i^2$ terms because they are quadratic in perturbation theory we get:

$$\frac{\partial^2_{(0)}\phi^{IR}}{\partial N^2} + \left(3 - \frac{1}{2M_{PL}^2}\left(\frac{\partial_{(0)}\phi^{IR}}{\partial N}\right)^2\right)\frac{\partial_{(0)}\phi^{IR}}{\partial N} + \left(3M_{PL}^2 - \frac{1}{2}\left(\frac{\partial_{(0)}\phi^{IR}}{\partial N}\right)^2\right)\frac{V_\phi\left(_{(0)}\phi^{IR}\right)}{V\left(_{(0)}\phi^{IR}\right)}$$

$$-\frac{\partial\xi_1}{\partial N} - \xi_2 - \left[3 - \frac{1}{2M_{PL}^2}\left(\frac{\partial_{(0)}\phi^{IR}}{\partial N}\right)^2 - \frac{1}{M_{PL}^2}\left(\frac{\partial_{(0)}\phi^{IR}}{\partial N}\right)^2 - \frac{V_\phi\left(_{(0)}\phi^{IR}\right)}{V\left(_{(0)}\phi^{IR}\right)}\left(\frac{\partial_{(0)}\phi^{IR}}{\partial N}\right)\right]\xi_1, \qquad (112)$$

which can be conveniently written if we use an auxiliary variable $_{(0)}\pi^{IR}$:

$$_{(0)}\pi^{IR} = \frac{\partial_{(0)}\phi^{IR}}{\partial N} + \xi_1,$$

$$\frac{\partial_{(0)}\pi^{IR}}{\partial N} + \left(3 - \frac{\left(_{(0)}\pi^{IR}\right)^2}{2M_{PL}^2}\right)_{(0)}\pi^{IR} + \left(3M_{PL}^2 - \frac{\left(_{(0)}\pi^{IR}\right)^2}{2}\right)\frac{V_\phi\left(_{(0)}\phi^{IR}\right)}{V\left(_{(0)}\phi^{IR}\right)} = -\xi_2. \qquad (113)$$

### 6.1.1. Characterization of the Noises

The interpretation of $\xi_1$ and $\xi_2$ (note that $\xi_3$ no longer appears in the final equation of motion) as classical noises is not trivial because they are, strictly speaking, quantum operators. In order to see how they are effectively classical, we can compute the two-point correlation function of $\xi_1$ for example at equal space point, the result is:

$$\langle 0|\xi_1(N_1)\xi_1(N_2)|0\rangle = \frac{\partial}{\partial N}\left(\sigma a \frac{H^b}{{}_{(0)}\alpha^{IR}}\right)\left(\sigma a \frac{H^b}{{}_{(0)}\alpha^{IR}}\right)^2\left|\delta\phi(N_1)_{k=\left(\sigma a \frac{H^b}{{}_{(0)}\alpha^{IR}}\right)}\right|^2\delta(N_1-N_2),\tag{114}$$

where we have used (104) together with the commutation relation (48). From (114) we see that $\delta\phi_{\mathbf{k}}$, which is the solution for the field perturbation over the local patch of size $\left(\sigma a \frac{H^b}{{}_{(0)}\alpha^{IR}}\right)^{-1}$, is evaluated at the coarse-grained scale, i.e., well outside the Hubble horizon. It can then be shown that at those scales, any perturbation that started from a coherent vacuum state has evolved into a highly squeezed state [119,120], which means that we can consider $\left|\delta\phi(N)_{k=\left(\sigma a \frac{H^b}{{}_{(0)}\alpha^{IR}}\right)}\right|^2$ as the power spectrum of a classical random variable, whose time evolution is consistent with probabilities conserved along classical trajectories.

Once this is clarified, we are now in position to describe $\xi_1$ as a classical white noise (Its "white" nature is due to the presence of $\delta(N_1-N_2)$ in the two-point correlator. Note that this is a consequence of the the choice of the Heaviside theta function as Window function, any other choice would lead to coloured noises, which are much more difficult to deal with, both analytically and numerically) with variance given in (114). Furthermore, since the field fluctuations are Gaussian to a good level of approximation, the variance computed in (114) is enough to fully characterize $\xi_1$. Finally, in order to characterize the system (113) we also need:

$$\langle 0|\xi_1(N_1)\xi_2(N_2)|0\rangle = \langle 0|\xi_2(N_1)\xi_1(N_2)|0\rangle^* =$$

$$\frac{\partial}{\partial N}\left(\sigma a \frac{H^b}{{}_{(0)}\alpha^{IR}}\right)\left(\sigma a \frac{H^b}{{}_{(0)}\alpha^{IR}}\right)^2\left(\delta\phi(N_1)_{k=\left(\sigma a \frac{H^b}{{}_{(0)}\alpha^{IR}}\right)}\frac{\partial\delta\phi^*(N_1)_{k=\left(\sigma a \frac{H^b}{{}_{(0)}\alpha^{IR}}\right)}}{\partial N}\right)\delta(N_1-N_2),$$

$$\langle 0|\xi_1(N_1)\xi_1(N_2)|0\rangle = \frac{\partial}{\partial N}\left(\sigma a \frac{H^b}{{}_{(0)}\alpha^{IR}}\right)\left(\sigma a \frac{H^b}{{}_{(0)}\alpha^{IR}}\right)^2\left|\frac{\partial\delta\phi(N_1)_{k=\left(\sigma a \frac{H^b}{{}_{(0)}\alpha^{IR}}\right)}}{\partial N}\right|^2\delta(N_1-N_2).\tag{115}$$

The characterization of $\xi_1$ and $\xi_2$ as white noises give us now a intuitive picture of the physics behind the stochastic formalism. As explained before, different functions for ${}_{(0)}\alpha^{IR}$ and ${}_{(0)}\phi^{IR}$ (in uniform-N gauge) describe the evolution of different FLRW patches, the way of getting these different functions is now clear if we see $\xi_1$ and $\xi_2$ as random variables. For example, the evolution of a specific patch, let us call it ${}^1FLRW$, will be given by ${}^1_{(0)}\alpha^{IR}$ and ${}^1_{(0)}\phi^{IR}$, whose specific form will be determined by the random values that the noises ${}^1\xi_1$ and ${}^1\xi_2$ will pick at each time step. Now, if we want to describe a second patch ${}^2FLRW$, we just have to solve again the stochastic equation with different random values for the noises ${}^2\xi_1$ and ${}^2\xi_2$, always satisfying the statistics described by (114) and (115). Like this, we are then able to describe the evolution of an ensemble of FLRW patches by solving many times the same stochastic equation with different random values for the noises. The correlators between these patches are then simply described by statistical moments of the IR variables.

The stochastic system is already fully characterized by (113)–(115); however, this system is very difficult to solve because, in order to compute the variance of the noises $\xi_1$ and $\xi_2$, we need to solve the perturbation equations and compute $\delta\phi_{\mathbf{k}}$ over a stochastic background every time step, which makes the systen non-Markovian. Although there are numerical algorithms capable of doing so [67], it would be very convenient if we were able to write an analytical solution for $\delta\phi_{\mathbf{k}}$ in terms of the *IR* variables, that would make the system Markovian and easily solvable. Let us try to solve the equation for $\delta\phi_{\mathbf{k}}$ over the stochastic local background.

We know that, as a consequence of the separate universe approach used here, the linearized equation for the field perturbation in the uniform-N gauge will be the same as the equation for the "separate-universe" gauge invariant variable $Q^{sep}$ defined in (68), in other words, $Q^{sep} = \delta\phi_{\mathrm{f}} = \delta\phi_{\delta N}$ under the separate universe assumption. Now, since the linearized equation for $Q^{sep}$ only gives the correct solution for the true gauge-invariant MS variable $Q$ defined in (44) if we set $\epsilon_1 = 0$ as checked in Section 4.1.2, the equation we will try to solve here is the linearized equation for the gauge invariant quantity $Q$ over a local background in which all the SR parameters have been set to zero (note that, in order for this approximation to work we need $-\frac{3}{2}\epsilon_2 - \frac{1}{4}\epsilon_2^2 \sim \mathcal{O}(\epsilon_1)$, which is true for SR and USR. This is because we are setting the combination of SR parameters multiplying $Q$ in (45) to zero), in other words, we will try to solve the equation for $Q$ as if the local background were an exact de-Sitter, which is:

$$\frac{\partial^2 \delta\phi_{\mathbf{k}}}{\partial N^2} + 3\frac{\partial \delta\phi_{\mathbf{k}}}{\partial N} + \left({}_{(0)}\alpha^{IR}\right)^2 \frac{k^2}{a^2 \left(H^b\right)^2}\delta\phi_{\mathbf{k}} = 0, \tag{116}$$

or, using the Hamiltonian constraint of (109):

$$\frac{\partial^2 \delta\phi_{\mathbf{k}}}{\partial N^2} + 3\frac{\partial \delta\phi_{\mathbf{k}}}{\partial N} + \frac{k^2}{a^2}\left[\frac{3M_{PL}^2 - \frac{\left({}_{(0)}\pi^{IR}\right)^2}{2}}{V\left({}_{(0)}\phi^{IR}\right)}\right]\delta\phi_{\mathbf{k}} = 0. \tag{117}$$

Now, both the functions ${}_{(0)}\pi^{IR}$ and $V\left({}_{(0)}\phi^{IR}\right)$ are stochastic so, in order to know them we must already know the value for the variable $\xi_1$. This makes Equation (117) not analitically solvable, more concretely, the stochastic system of (113) is non-Markovian, meaning that the value of the noises at each time $N$ will depend on the whole evolution of the stochastic patch up to $N$.

The only option remaining to solve the stochastic system of (113) seems then to be numerically; however, a further and very important approximation is usually done, which consists on assuming that the IR (and stochastic) quantities do not differ much from their global background (and deterministic) counterpart, namely $Y^{IR}X^{UV} \simeq Y^b X^{UV} + \mathcal{O}\left(\left(X^{UV}\right)^2\right)$. Here $X^{UV}$ and $Y^{IR}$ are any UV and IR functions, we then define $Y^b$ as the equivalent background function of $Y^{IR}$. Under this approximation, the last term of (116) is:

$$k^2 \left(\frac{{}_{(0)}\alpha^{IR}}{aH^b}\right)^2 \delta\phi_{\mathbf{k}} \simeq \left(\frac{k}{aH^b}\right)^2 \delta\phi_{\mathbf{k}} + \mathcal{O}\left(\delta\phi_{\mathbf{k}}^2\right), \tag{118}$$

where we have substituted ${}_{(0)}\alpha^{IR}$ by its background value, i.e., 1. Under this approximation we can write a analytical solution for (116), that, once evaluated at coarse-grained scale, will correspond to the long-wavelength limit of solution (51) with $\nu = \frac{3}{2}$, i.e.,

$$\left|\delta\phi(N)_{k=(\sigma aH^b)}\right|^2 = \frac{\left(H^b\right)^2}{2(\sigma aH^b)^3}. \tag{119}$$

and therefore the correlators (114) and (115) can be written as follows:

$$\langle \xi_1(N_1)\xi_1(N_2)\rangle = \left(\frac{H^b}{2\pi}\right)^2 \delta(N_1 - N_2)\,, \tag{120}$$

$$\langle \xi_1(N_1)\xi_2(N_2)\rangle = \langle \xi_2(N_1)\xi_2(N_2)\rangle = 0\,, \tag{121}$$

Under this approximation, the stochastic system of (113) simplifies considerably and becomes a Markovian process with additive noises, meaning that $\xi_1$ depends only on time, which is true as long as we are using the global background to characterize it. The system is:

$$_{(0)}\pi^{IR} = \frac{\partial\, _{(0)}\phi^{IR}}{\partial N} + \frac{H^b}{2\pi}\xi(N)\,,$$

$$\frac{\partial\, _{(0)}\pi^{IR}}{\partial N} + 3\, _{(0)}\pi^{IR} + M_{PL}^2 \frac{V_\phi\left(_{(0)}\phi^{IR}\right)}{V\left(_{(0)}\phi^{IR}\right)} = 0\,, \tag{122}$$

where $\langle \xi(N_1)\xi(N_2)\rangle = \delta(N_1 - N_2)$ and we have dropped $\mathcal{O}(\epsilon_1)$ terms in order to be consistent with the computation of the noises and the reasoning above (116). Note that in SR the acceleration of the field is also of higher order in SR parameters so in this case the stochastic equation would simplify even more:

$$\frac{\partial\, _{(0)}\phi^{IR}}{\partial N} + M_{PL}^2 \frac{V_\phi\left(_{(0)}\phi^{IR}\right)}{V\left(_{(0)}\phi^{IR}\right)} = -\frac{H^b}{2\pi}\xi(N)\,. \tag{123}$$

Although the approximation used in order to arrive to (122) seems very useful, it has important consequences, in fact, it is equivalent to state that any $Y^{IR} - Y^b \sim \mathcal{O}\left(X^{UV}\right)$. Thus, we immediately see that if this approximation holds, the system in (122) can only reproduce the results of linear theory at leading order in SR parameters, and hence it does not give any non-perturbative (or even non-linear) information. In fact, (122) is slightly inconsistent. The point is that, by the same approximation adopted on the right hand side, the left hand side should also be linearized (in fact the linearization of the left-hand side of (122) is sometimes performed when recursive methods are used in order to solve the stochastic system as it can be seen in [42–44]). This inconsistency can however give some information when comparing the results from the stochastic formalism with the ones of linear perturbation theory, in fact, since we know that the correlations functions calculated with the stochastic system of (122) will coincide, up to second order in perturbation theory, to the ones calculated in linear perturbation theory with QFT methods, any inconsistency between the two approaches will signal the break-down of perturbation theory.

Since the stochastic Equation (116) and its deterministic counterpart with $_{(0)}\alpha^{IR} = 1$ have the same structure, one could be tempted to write the solution of (116) in a similar way as the solution (119) but substituting $H^b$ by $\frac{H^b}{_{(0}\alpha^{IR}}$, i.e:

$$\left|\delta\phi(N)_{k=\left(\sigma a\frac{H^b}{_{(0}\alpha^{IR}}\right)}\right|^2 = \left(\frac{H^b}{_{(0}\alpha^{IR}}\right)^2 \frac{1}{2\left(\sigma a\frac{H^b}{_{(0}\alpha^{IR}}\right)^3}\,, \tag{124}$$

and hence the stochastic system of (113) would approximately be:

$$_{(0)}\pi^{IR} = \frac{\partial_{(0)}\phi^{IR}}{\partial N} + \frac{H^b}{2_{(0)}\alpha^{IR}\pi}\xi(N),$$

$$\frac{\partial_{(0)}\pi^{IR}}{\partial N} + 3_{(0)}\pi^{IR} + 3M_{PL}^2\frac{V_\phi\left(_{(0)}\phi^{IR}\right)}{V\left(_{(0)}\phi^{IR}\right)} = 0. \tag{125}$$

The system of (125) represents a Markovian process with non-additive noises (the term proportional to $\xi$ does no longer depends only on time), which is, provided that (124) holds, able to describe the inhomogeneities generated during inflation in a non-perturbative way. Unfortunately, we cannot trivially generalize the solution (119) for a deterministic equation to a solution (124) for a stochastic equation, this is due to the differences between stochastic and deterministic integrals (see for example [121]). This is why, although it is the most common approach found in the literature, we will not trust (125) to describe non-linear inflationary effects in this review.

Having discarded this option, we are left with three different ways of solving the stochastic system of (113):

1. We can use the system (113) where the noises are computed numerically over the stochastic local background, for example, using the algorithm described in [67].

   - **Pros:** It describe non-linear inflationary dynamics.
   - **Cons:** It is very difficult to solve due to the non-Markovianity of the process. Furthermore, it is only valid up to leading order in SR parameters due to the use of the separate universe approach.

2. We can use the system of (122)

   - **Pros:** It is a Markovian process with additive noises for which even analytical solutions can be obtained (see Section VI of [65]).
   - **Cons:** It only describes linear perturbations whenever they are approximately described by solution (119), which is not the case during some interesting regimes for PBH formation, such as a SR-USR transition where $\nu \neq \frac{3}{2}$. It is then even less precise than the linear separate universe approach.

3. Finally, we can solve the MS equation over a global background at all orders in SR parameters, i.e., we can solve (45) and characterize the noises as in (114) with this solution. This can be done because, as we have already indicated, under the separate universe assumption $Q = \delta\phi_{\delta N}$. Once the noises are characterized in this way, we can use the stochastic system (113) and solve the dynamics.

   - **Pros:** It is a Markovian process with additive noises able to describe the linear dynamics of the inflationary perturbations even when they are not approximately described by (119).
   - **Cons:** It is not capable of describing any non-linear effects and it is inconsistent generically at leading order in $\epsilon_1$ due to the use of the separate universe approach. This inconsistency can be clearly seen in the term proportional to $\xi_1$ in (112), which in the background can be written as $\left(3 - \epsilon_1 + \frac{\epsilon_1\epsilon_2}{3-\epsilon_1}\right)$ and not $(3 - \epsilon_1)$ as it should be if it came from the MS equation. In fact $\left(3 - \epsilon_1 + \frac{\epsilon_1\epsilon_2}{3-\epsilon_1}\right)$ is precisely the term that appears in the equation for $Q^{sep}$ (69) derived in Section 4.1.2, making it clear that this inconsistency is a consequence of the separate universe approach.

Due to the problems that the stochastic formalism presented in this section has (namely its validity only up to leading order in $\epsilon_1$ due to the separate universe assumption and its difficulty to describe non-linear dynamics), it does not seem very promising if we want to describe the non-perturbative dynamics of the inohomogeneities generated during inflationary regimes of interest for PBH formation with enough precision. However, in the

next subsection we will present a stochastic formalism (firstly presented in [65]) which is at least able to reproduce linear perturbation theory at all orders in SR parameters, and with the potential to correctly describe the non-perturbative dynamics of the inhomogeneities during any inflationary regime.

However, before doing so, and in order to finish this subsection, it is important to know that the stochastic system of (113) can be straightforwardly derived by splitting into a IR and an UV only the scalar field in the separate universe equations of (64). In this case we would have used the metric (63) for the local patch instead of the metric (83) as we have done here. We have not chosen this option in this review because of three main reasons:

1.  It does not make it clear that we have used gradient expansion, and hence the problem of not using the momentum constraint that we solve in the next subsection is difficult to remark.
2.  Since it only works if the time variable is unperturbed, it could lead us to think that the number of e-folds $N$ is the only allowed time variable for a stochastic formalism that describes all the scalar inhomogeneities in terms of the inflaton field. On the contrary, the derivation used in this review is valid for any time variable and makes it clear that the description of inhomogeneities in terms solely of the inflaton field is only a gauge choice.
3.  It does not explicitly obtains the linear equations used for the characterization of $\delta\phi_{\mathbf{k}}$, more concretely, for example this derivation would miss the $\frac{k^2}{a^2}\boldsymbol{\alpha}_{\mathbf{k}}^{UV}$ term that multiplies the Heaviside theta in (105).

*6.2. Stochastic Formalism Based on $\mathcal{O}(\sigma^0)$ Gradient Expansion*

As many times claimed during the review, the separate universe approach generically fails to give the correct long-wavelength evolution of the inhomogeneities at $\mathcal{O}(\epsilon_1)$. The $\mathcal{O}(\sigma^0)$ gradient expansion solves this problem by including both non-local terms and the momentum constraint, this is why in this section we will construct a stochastic formalism based on $\mathcal{O}(\sigma^0)$ gradient expansion.

First of all, it is important to remark that some of the affirmations we did about the uniform-N gauge at the beginning of Section 6.1 are no longer correct, more concretely, we cannot longer study the scalar inhomogeneities in terms solely of the inflaton field. This is clear form linear perturbation theory where the MS variable can be written as

$$Q = \delta\phi_{\mathrm{f}} = \delta\phi_{\delta N} - \frac{\partial\phi^b}{\partial N}\frac{1}{3}\nabla^2 E_{\delta N}\,, \tag{126}$$

so if we insist on using the uniform-N gauge we must also take into account the contribution from $E$ when studying scalar perturbations.

We could also use spatially flat gauge in this case and forget about E; however, in this case we should take into account all the terms proportional to $\left(\,_{(0)}\beta_{\mathrm{f}}\right)^i$ that appear for example in (90), this is why we will keep using the uniform-N gauge, where $\beta^i = 0$.

One can easily check that the stochastic equations for the evolution of the extrinsic curvature (106), for the Hamiltonian constraint (109) and for the evolution of the field (110) do not change when including $E$ and hence the stochastic system is still the one given by (113). The only difference will then be given by the inclusion of the momentum constraint (15), which, in uniform-N gauge can be written as:

$$D^j\left(-\frac{H^b}{2\alpha}\frac{\partial\tilde{\gamma}_{ij}}{\partial N}\right) - \frac{2}{3}D_i K = -\frac{1}{M_{PL}^2\alpha}\frac{\partial\phi}{\partial N}\partial_i\phi\,, \tag{127}$$

where we have used the evolution Equation for $\tilde{\gamma}_{ij}$ (18) to eliminate $\tilde{A}_{ij}$. Splitting (127) into IR and UV and using the decomposition of $\tilde{\gamma}_{ij}$ explained around (87) to keep only $\mathcal{O}(\sigma)$ terms in the IR part (remember that the $\mathcal{O}(\sigma^0)$ information from the momentum constraint

can only be extracted if we write the momentum constraint up to $\mathcal{O}(\sigma)$), we can write the stochastic equation for the momentum constraint:

$$_{(0)}\partial_i\left(\frac{\partial}{\partial N}\left(\frac{1}{3}\nabla^2 C^{IR}\right)\right) - \frac{_{(0)}\partial_i\alpha^{IR}}{_{(0)}\alpha^{IR}} + \frac{\partial\,_{(0)}\phi^{IR}}{\partial N}\frac{_{(0)}\partial_i\phi}{2M_{PL}^2} = -\partial_i\xi_4\,, \tag{128}$$

where $\xi_4$ is defined similarly to $\xi_1$, $\xi_2$ and $\xi_3$, i.e.,

$$\xi_2 \equiv -\frac{\partial}{\partial N}\left(\sigma a\frac{H^b}{_{(0)}\alpha^{IR}}\right)\int\frac{d\mathbf{k}}{(2\pi)^{3/2}}\delta\left(k - \sigma a\frac{H^b}{_{(0)}\alpha^{IR}}\right)\left(-\frac{k^2}{3}\mathcal{C}_{\mathbf{k}}^{UV}\right). \tag{129}$$

With the addition of the stochastic Equation (128) to the system of (113) obtained before, we have a stochastic formalism able to describe the non-linear evolution of scalar inhomogeneities at all orders in SR parameters.

However, it is not all good news: firstly, since the construction of the gradient expansion in Section 5, we have been neglecting possible interactions scalar-tensor or scalar-vector, reason why the stochastic formalism constructed here will not take these interactions into account either. Secondly, we do not exactly know how to extract the $\mathcal{O}(\sigma^0)$ information form (128) in a fully non linear way. Finally, we do not know which is the combination of $_{(0)}\phi^{IR}$ and $\nabla^2 C^{IR}$ that give us the correct and non-perturbative and gauge invariant quantity that describe scalar inhomogeneities, i.e., we do not have a non-linear generalization of the MS variable. The first issue is beyond the scope of this review. With respect to the third one, it is true that a non-linear gauge invariant variable at leading order in gradient expansion has been defined in [117,118] as:

$$\partial_i Q^{NL} \equiv \partial_i\phi + \frac{1}{\alpha}\frac{\partial\phi}{\partial N}\partial_i\zeta$$

However, the variable above does not include the term proportional to $\nabla^2 E$ in its linearization. Reason why it can be only interpreted as a non-linear generalization of $Q^{sep}$. This is the reason we will not use the non-linear variable defined above and we will solve the second and third issues, at least approximately, by imposing that both the momentum constraint and the non-linear generalization of the MS variable match their linear counterpart when the global background is subtracted. In this way, the $\mathcal{O}(\sigma^0)$ information of (128) can be straightforwardly extracted and the whole system of stochastic equations based on the $\mathcal{O}(\sigma^0)$ gradient expansion and hence valid at all orders in SR parameters is:

$$_{(0)}\pi^{IR} = \frac{\partial\,_{(0)}\phi^{IR}}{\partial N} + \xi_1\,,$$

$$\frac{\partial\,_{(0)}\pi^{IR}}{\partial N} + \left(3 - \frac{\left(_{(0)}\pi^{IR}\right)^2}{2M_{PL}^2}\right)_{(0)}\pi^{IR} + \left(3M_{PL}^2 - \frac{\left(_{(0)}\pi^{IR}\right)^2}{2}\right)\frac{V_\phi\left(_{(0)}\phi^{IR}\right)}{V\left(_{(0)}\phi^{IR}\right)} = -\xi_2\,,$$

$$\frac{\partial}{\partial N}\left(\frac{1}{3}\,_{(0)}\nabla^2 C^{IR}\right) - \left(H^b\sqrt{\frac{3M_{PL}^2 - \frac{\left(_{(0)}\pi^{IR}\right)^2}{2}}{V\left(_{(0)}\phi^{IR}\right)}} - 1\right) + \frac{1}{2_{PL}^2}\frac{\partial\,_{(0)}\phi^{IR}}{\partial N}\left(_{(0)}\phi^{IR} - \phi^b\right) = -\xi_4\,, \tag{130}$$

where we have used the Hamiltonian constraint to eliminate $_{(0)}\alpha^{IR}$ in the last equation. Note that $\xi_1,\xi_2$ and $\xi_4$ are constructed in the uniform-N gauge, which is no longer equivalent to the spatially flat gauge.

To see the gauge transformation between spatially flat and uniform-N gauges in linear theory one can see [60], where it is claimed that the differences between those two gauges is always of higher order in gradient expansion; however, this conclusion is reached by

considering the value of $\epsilon_1$ at horizon crossing ($\epsilon_1^*$ there) to be constant with $k$, which is generically not a good approximation beyond SR. In fact in [67] it is shown numerically that the difference between $\delta\phi_f$ and $\delta\phi_{\delta N}$ can be $\mathcal{O}(\epsilon_1)$ in regimes of interest for PBH formation, in agreement with the differences between the separate universe approach and $\mathcal{O}(\sigma^0)$ gradient expansion remarked in this paper.

Finally, as suggested in [122–124], we can define the non-linear counterpart of the MS variable of (44) at leading order in gradient expansion as:

$$Q^{IR} = {}_{(0)}\phi^{IR} - \phi^b - \frac{\partial {}_{(0)}\phi^{IR}}{\partial N}\frac{1}{3}{}_{(0)}\nabla^2 C^{IR}, \tag{131}$$

where we remind the reader that ${}_{(0)}\phi^{IR}$ and ${}_{(0)}\nabla^2 C^{IR}$ are both in the uniform N gauge.

The stochastic formalism based on the $\mathcal{O}(\sigma^0)$ gradient expansion presented in this section was introduced for the first time, although in a different gauge, in [65], where the authors show numerically that, when computing the noises over a global background, i.e., when using the approximation described around (118), the two point correlator computed using the stochastic formalism perfectly matches with the one computed in linear perturbation theory at all orders in SR parameters, as expected. This already represents an important improvement with respect to any stochastic formalism based on the separate universe approach as the one explained in Section 6.1.

### 6.3. Stochastic Formalism Versus Linear Perturbation Theory

Once the stochastic formalism has been properly introduced, the simplest way to know if there exists any important non-perturbative effect during inflation is to compare the results obtained within the stochastic framework with the results coming from linear perturbation theory explained in Section 4. In order to do so, it is crucial to realize that the results coming from the stochastic formalism are in real space whereas the result for the linear MS variable in terms of Hankel function of (51) is in Fourier space. Due to the difficulty when trying to express stochastic correlators in Fourier space, we will compare the stochastic two-point correlator with the linear two-point correlator, both in real space.

As explained below (115), the correlators between different FLRW patches are described by statistical moments of IR variables, so the two point correlator of the scalar inhomogeneities is:

$$\langle {}_{(0)}\phi^{IR} {}_{(0)}\phi^{IR}\rangle = Var\left({}_{(0)}\phi^{IR}\right), \tag{132}$$

if we are using the separate universe approach or

$$\langle Q^{IR} Q^{IR}\rangle = Var\left(Q^{IR}\right), \tag{133}$$

if we are using $\mathcal{O}(\sigma^0)$ in gradient expansion.

On the other hand, the two-point correlator in linear perturbation theory is defined as (see (52)):

$$\langle Q(N)Q(N)\rangle = \int_{\sigma a(N=0)H^b(N=0)}^{\sigma a(N)H^b(N)} \frac{dk}{k}\mathcal{P}_Q(k,N), \tag{134}$$

where we are now using the power spectrum evaluated at the same spatial point

$$\mathcal{P}_Q(k,N) \equiv \frac{k^3}{2\pi^2}|Q_{\mathbf{k}}(N)|^2, \tag{135}$$

where $Q_{\mathbf{k}}$ in uniform-N gauge is given by (126). The limits of (134) correspond to the selection of modes inside the coarse-grained scale defined by $k = \sigma a(N)H^b(N)$ and they are necessary to correctly compare the two point correlators obtained in both formalisms, in fact, in the stochastic formalism the IR part of the field recieves stochastic kicks from $N = 0$ onwards. Thus the first k-mode from which the IR field recieves a kick is the one with $k = \sigma a(N = 0)H^b(N = 0)$.

Whenever $\mathcal{P}_Q(k, N)$ does not depend on $N$, one can do a very useful approximation, which consist on evaluating the power spectrum at coarse-grained scale crossing, i.e., at $k = \sigma a H^b$, and assume that this value does not change with time. This would allow us to write (134) as:

$$\langle Q(N)Q(N) \rangle = \int_0^N \mathcal{P}\Big(k = \sigma a(N')H^b(N')\Big)dN', \tag{136}$$

In this case one could write the power spectrum as the derivative with respect to the number of e-folds $N$ of the correlator in real space.

$$\mathcal{P}_Q(k) = \frac{d}{dN}\langle Q(N)Q(N) \rangle. \tag{137}$$

However, and although it has been sometimes wrongly used for a time-dependent power spectrum [33], this technique cannot be used if the power spectrum evolves with time, which makes it only valid at zeroth order in $\mathcal{O}(\epsilon_1)$.

Finally, and due to the difficulty in the definition of non-linear gauge invariant variables (remember for example that the "trivial" non-linear generalization of the MS variable of (131) is not guaranteed to be the correct one), some authors have used the so-called stochastic-$\delta N$ formalism to compute the non-linear curvature perturbation, which is very useful when dealing with PBHs, by studying the statistics in the number of e-folds [36,41,46,50,51]. However, and similarly to what happens with the usual $\delta N$ formalism, the stochastic-$\delta N$ formalism uses the separate universe approach and hence it is generically only valid at leading order in SR parameters.

## 7. Conclusions

The stochastic approach to inflation seems the most promising way to study, in a non-perturbative way, the probability distribution of the scalar inhomogeneities generated during inflation and responsible for the creation of PBHs. However, as remarked during the review, there is still a lot of work to do in that direction. Indeed, although very useful, the most common and widely used stochastic formalism has some difficult problems to solve, which can be summed up in three points:

1. As we have seen during the review, the stochastic formalism uses a gradient expansion for the IR part and a perturbative expansion for the UV part in such a way that the IR part, due to the large wavelength of the characteristic inhomogeneities that form this sector, can be described as a local FLRW universe. This description, that we called $\mathcal{O}(\sigma^0)$ gradient expansion, relates the different local FLRW patches via non-local terms and the momentum constraint, and describes at all orders in SR parameters the correct dynamics of long wavelength scalar inhomogeneities. However, it presents some problems such as the extraction of the $\mathcal{O}(\sigma^0)$ information from the momentum constraint.

   This is the reason why it is usually assumed that the different local FLRW universes evolve independently from each other, which is an assumption known as the separate universe approach. Under this approximation, the problem with the momentum constraint disappears (the momentum constraint itself disappears); nevertheless, we checked in Section 4.1.2 that this assumption fails to describe the correct long-wavelength dynamics of scalar perturbations generically at $\mathcal{O}(\epsilon_1)$ already in its linear limit. For this reason, the stochastic formalism commonly used presented in Section 6.1, which is based on the separate universe approach will fail to describe the non-perturbative dynamics of the scalar inhomogeneities at $\mathcal{O}(\epsilon_1)$.

   A stochastic formalism that does not uses the separate universe approach can also be constructed, as we did in Section 6.2. However, this option is not without its difficulties, more concretely, it has problems when extracting the long wavelength information from the momentum constraint and when describing the scalar inhomogeneities with a gauge invariant generalization of the MS variable.

2. On the other hand, the UV part of the inhomogeneities is assumed to behave perturbatively, having as a background the local FLRW background defined by the IR part. Since the UV part acts as a stochastic noise for the IR part and the IR part is necessary to characterize the UV noises, the stochastic approach to inflation is generically a non-Markovian process, meaning that the value of the noises depend on the whole history of the local patch over which they are computed.

   This does not represent a huge problem when solving the stochastic equations numerically; however, in order to have analytical results we need to do some other approximation that makes the process Markovian. This approximation consists of computing the stochastic noises over the global background instead of over the local one. Unfortunately, doing so is equivalent to assume that all the IR inhomogeneities are linear in perturbation theory.

3. Finally, and although we have not paid too much attention to this problem, any stochastic approach that aims to describe the non-perturbative behaviour of scalar inhomogeneities at the long wavelength limit should also take into account scalar-vector and scalar-tensor interactions, which no longer decouple beyond linear perturbation theory.

**Funding:** This research received no external funding.

**Acknowledgments:** I would like to thank Cristiano Germani for our weekly discussions and his constant support and encouragement. I would also like to thank Misao Sasaki, Vincent Vennin and Danilo Artigas for the many correspondences and help in understanding the gradient expansion method and the stochastic formalism. I am supported by the Spanish MECD fellowship PRE2018-086135. The group I belong to is also supported by the Unidad de Excelencia Maria de Maeztu Grants No. MDM-2014-0369 and CEX2019-000918-M, and the Spanish national grants FPA2016-76005-C2-2-P, PID2019-105614GB-C22 and PID2019-106515GB-I00.

**Conflicts of Interest:** The authors declare no conflict of interest.

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
