# Peer review of "Review on Stochastic Approach to Inflation"

_universe, doi:10.3390/universe8060334_

Round 1
Reviewer 1 Report
Formulations for calculating fluctuations in the early Universe are discussed in detail. This review is likely to be useful for some interested in the topic, so I recommend publication, but I have a few remarks. English should be improved throughout the text before publication. The author emphasizes PBHs at first, for instance including it in the title, but PBH formation is not discussed that much, so it seemed a bit strange.
Finally, about the following statement:
However, to statistically generate 61 enough PBHs for this to hold one needs, at least, a power spectrum of primordial curvature 62 perturbations several order of magnitudes larger than the one observed in the cosmic 63 microwave background (CMB).
This statement appears misleading since enhancement of power spectrum is only one possibility, one could think of other mechanisms of PBH formation which does not use primordial fluctuations, or one could think of non-Gaussianity of fluctuations to make PBHs, even without enhancing the power spectrum.
Author Response
Dear referee,
Thank you very much about your comments,
I have tried to improve English throughout the text as you requested. Regarding your comment about the title, you are right, I did not discuss a lot about PBH so I decided to slightly modify the title.
Finally, regarding your comment about the statement of the generation of PBH, you are again right, I have added a footnote at the beginning of section 2 clarifying the point you arised.
Best regards,
Diego.
Reviewer 2 Report
The report is attached as a PDF file.

Author Response
Dear referee
Thank you very much for your comments.
You are right both saying that the review was lacking a bit of historical context about stochastic inflation and also saying that PBH formation is not discussed at all.
I have included a detailed historical analyisis of stochastic inflation in the introduction where I have added some new references. This summary of the long history behind stochastic inflation is also very useful in order to understand the way the review is structured.
Regarding the issue about PBH formation, I have modified the title of the review so it now describes better the content of the review.
I hope that with these modifications the review can now be recommended for publication.
Best regards,
Diego.
Reviewer 3 Report
In this paper, the author presented a review on the state-of-the-art the mathematical framework known as stochastic inflation, paying special attention to its derivation and giving references for the readers interested in results coming from the application of the stochastic framework to different inflationary scenarios, especially to those of interest for primordial black hole formation. During the derivation of the stochastic formalism, the author emphasized two aspects in particular: the difference between the separate universe approach and the true long-wavelength limit of scalar inhomogeneities and the generically non-Markovian nature of the noises that appear in the stochastic equations.
The results are correct. The work is well motivated and well-grounded from observational viewpoints. The calculations are also correct. I would like to accept this paper as is for publication in Universe.
Author Response
Dear Referee,
Thank you very much for your comments,
Best regards,
Diego.
Reviewer 4 Report
The paper contains a rather useful and critical analysis of the stochastic approach to inflation, its predictions for spectrum of density fluctuations and their possible links to PBH formation. However, in the present form the paper cannot be recommended for publication and should be reconsidered after major revision, which puts the discussion in the general framework of PBH probes for the physics of very early Universe (the references to the reviews by A.Dolgov and M.Khlopov should be inevitably addressed, or at least mentioned in the Bibliography) , specifying the physical meaning of the considered formalism and its problems. It is also necessary to put all the details of calculations to appendices making the main text more comprehensive and clear for the Universe reader.
Author Response
Dear referee,
Thank you very much for your comments,
I agree with you that there is not a lot of discussion on how PBH can probe the physics of the very early universe. Although this was not the original idea of the review, it is true that the title may confuse the reader about the importance of PBH in the review, so I decided to change it. I have also added a footnote at the beginning of section 4.2. with the references you indicated for the reader interested in PBHs as probes of the physics of the very early universe.
Regarding your comment about putting the details of the calculations to appendices. I think that this would made the review a little bit more obscure, the reason is that I think that every calculation written in the main text is necessary to understand the way the stochastic formalism is constructed so, in my opinion, putting these calculations in the appendix would lead to a less comprehensive review.
I hope that with these changes and comments the review can now be recommended for publication.
Best regards.
Diego.
Reviewer 5 Report
The author makes an interesting review about the theory of homogeneous inflation, study of anisotropies and subsequently about stochastic inflation which helps to analyse the inhomogeneities from a different perspective. It is clear the there is nothing new in the paper, but as a review it looks fine and it cites most of the most important research works developed previously. Before accepting the paper for publication, I would like to ask the author how can the stochastic formulation be accommodated if we analyse the inflationary scenario inside some modifications of gravity, like massive gravity for example. Please take a look at the following references for that purpose: 1). PTEP 2014 (2014) 023E02, where the perturbation theory was developed generically inside Massive gravity. Then how could we apply stochastic methods in such a case?
2). Phys. Rev. Lett. 106, 231101, (2010), where the general formulation of the theory of massive gravity was introduced.
The author could comment a little bit about this scenario. After the author takes into account these suggestions, the paper can be published.
Author Response
Dear referee,
Thank you very much for your comments,
Regarding your question about massive gravity, I am not an expert on that field but as far as I understand, massive gravity can be studied during inflation and it allows both a separation of scales between IR and UV modes and a perturbative expansion for the UV modes so I do not see any reason why stochastic formalism could not be applied in this context. I have added a paragraph at the beginning of section 6 clarifying this point.
I hope that with this modification the review can now be recommended for publication.
Best regards,
Diego.
Round 2
Reviewer 2 Report
Dear editor,
This is the second referee report on the manuscript entitled "Review on stochastic approach to inflation" by Diego Cruces. As an answer to my previous comments, the author has substantially updated the introduction and narrowed the topic down to the stochastic formalism of inflation. That is why I am happy to recommend it for publication.
Best regards,
The referee.
Reviewer 4 Report
The paper is improved after major revision and the revised version can be accepted for publication in the present form